

# Estimation of the height of turbulent mixing layer from data of Doppler lidar measurements using conical scanning by a probe beam

Viktor A. Banakh, Igor N. Smalikho, Andrey V. Falits

V.E. Zuev Institute of Atmospheric Optics SB RAS, Tomsk, Russia

*Correspondence to*: V.A. Banakh ([banakh@iao.ru](mailto:banakh@iao.ru))

**Abstract.** A method is proposed for determining the height of the turbulent mixing layer on the basis of the vertical profiles of the dissipation rate of turbulent energy, which is estimated from lidar measurements of the radial wind velocity using conical scanning by a probe beam around the vertical axis. The accuracy of the proposed method is discussed in detail. It is shown that for the estimation of the mixing layer height (MLH) with the acceptable relative error not exceeding 20%, the

signal-to-noise ratio should be no less than -16 dB, when the relative error of lidar estimation of the dissipation rate does not exceed 30%. The method was tested in an experiment in which the wind velocity turbulence was estimated in smog conditions due to forest fires in Siberia in 2019. The results of the experiment reveal that the relative error of determination of the MLH time series obtained by this method does not exceed 10% in the period of turbulence development. The estimates of the turbulent mixing layer height by the proposed method are in good agreement with the MLH estimated from

the distributions of the variance of radial velocity and the Richardson number in height and time.

## 1 Introduction

The turbulent mixing layer in the lower part of the Earth's atmosphere has an important role in the vertical transport of moisture, small gas constituents, and pollutants from the surface to the upper layers of the atmosphere. The turbulent mixing layer height is usually understood to be the thickness of the layer adjacent to the ground, in which incoming substances

become completely vertically distributed throughout the layer owing to convection or turbulence for an hour (Bonin et al., 2018). It is apparent that the higher the intensity of wind turbulence, the larger the mixing layer thickness.

There are different technical facilities that can be used for determining the mixing layer height. Doppler sodars, radioacoustic systems, and Doppler lidars are the most suitable for this task, as they allow for meteorological data used for turbulent parameter estimation to be measured in the atmospheric boundary layer (ABL) in real time with the required space

and time resolution (Bonin et al., 2018; Hogan et al., 2009; Tucker et al., 2009; Pichugina and Banta, 2010; Barlow et al., 2011; Helmis et al., 2012; Schween et al., 2014; Vakkari et al., 2015; Huang et al., 2017; Petenko et al., 2019). From the data of lidar measurements, the variance of radial velocity $\sigma_r^2(h)$, variances of vertical $\sigma_w^2(h)$ and horizontal $\sigma_u^2(h)$ and $\sigma_v^2(h)$ wind vector components, and turbulence kinetic energy (TKE) $E(h)$ can be estimated at different heights $h$.





In Bonin et al., 2018; Hogan et al., 2009; Tucker et al., 2009; Pichugina and Banta, 2010; Barlow et al., 2011; Schween et al., 2014; Vakkari et al., 2015; Huang et al., 2017, the mixing layer height (MLH) $h_{\mathrm{mix}}$ was determined from the decrease in the variance $\sigma_\alpha^2(h)$ ( $\alpha = r, w, u, v$ ) with height $h$ down to some minimum threshold value $\sigma_\alpha^2(h_{\mathrm{mix}}) = Thr_\alpha$ , at which the turbulence intensity becomes insufficient for efficient air mixing. The variances and MLH were estimated from pulsed

coherent Doppler lidar (PCDL) data through the use of various measurement strategies and data processing algorithms (Bonin et al., 2018; Hogan et al., 2009; Tucker et al., 2009; Pichugina and Banta, 2010; Barlow et al., 2011; Schween et al., 2014; Vakkari et al., 2015; Huang et al., 2017): (i) in the fixed strictly vertical direction of the probing beam, (ii) by vertical scanning, and (iii) by conical scanning by a beam around the vertical axis at a certain elevation angle $\varphi$ . In Bonin et al., 2018, the "composite fuzzy logic approach" based on the use of all three measurement geometries was applied to determine

$h_{\mathrm{mix}}$ .

According to analysis (Tucker et al., 2009), lidar measurements of the vertical profile of the variance of vertical velocity $\sigma_w^2(h)$ in the fixed vertical probing direction provide the best accuracy of estimation of the mixing layer height $h_{\mathrm{mix}}$ . However, it was shown (Bonin et al., 2018) that this is not always the case. In particular, during the propagation of internal gravity waves (IGWs), this method may significantly overestimate $h_{\mathrm{mix}}$ , and it becomes necessary to perform high-

frequency filtering of the data.

The turbulence energy dissipation rate, as well as variances of the fluctuations of wind vector components, characterizes the turbulence (air mixing) intensity and can also be used for the estimation of MLH $h_{\mathrm{mix}}$ . This was done for the first time in Vakkari et al., 2015, in which the diurnal profile of MLH $h_{\mathrm{mix}}(t)$ , where $t$ is time, was determined from the space-time distributions of the dissipation rate $\varepsilon(h,t)$ . The dissipation rate $\varepsilon(h,t)$ was estimated from temporal spectra of the vertical

velocity measured by lidar in the fixed strictly vertical direction of the probing beam with the use of the Taylor "frozen turbulence" hypothesis (O'Connor et al., 2010).

A method for estimating the turbulence energy dissipation rate $\varepsilon(h,t)$ from lidar data obtained with the use of conical scanning by a lidar probing beam around the vertical axis was developed in Banakh and Smalikho, 2013; Smalikho and Banakh, 2017. This measurement geometry does not require invoking the frozen turbulence hypothesis. In contrast to

O'Connor et al., 2010, this method (Banakh and Smalikho, 2013; Smalikho and Banakh, 2017) takes into account the spatial averaging of the radial velocity. The algorithm for calculating the error of the lidar estimation of the dissipation rate by this method (Banakh and Smalikho, 2013; Smalikho and Banakh, 2017) can be found in Banakh et al., 2017. It was shown in Smalikho and Banakh, 2017 that, in the case of moderate and strong turbulence and a sufficiently high signal-to-noise ratio $\mathrm{SNR}(h)$ , the accuracy of lidar estimates of the turbulence energy dissipation rate is, as a rule, markedly higher than the

accuracy of estimation of the variances of different wind vector components from lidar data. The estimate of the turbulence





energy dissipation rate $\varepsilon(h)$ remains reliable even during the appearance of IGWs with quite a high amplitude of harmonic oscillations of wind vector components (Banakh and Smalikho, 2018; Banakh et al., 2020).

In this paper, we report the results of estimating the turbulent mixing layer height from measurement data of pulsed coherent Doppler lidar obtained by the conical scanning by a probing beam. The mixing layer height $h_{\mathrm{mix}}(t)$ is determined

from the height–temporal distributions of lidar estimates of the turbulence energy dissipation rate. The accuracy of the obtained results is analyzed.

## 2 Method for determination of the turbulent mixing layer height from PCDL data obtained by conical scanning

It was shown in Smalikho and Banakh, 2017 that PCDL data obtained with the use of conical scanning by a probing beam around the vertical axis under the elevation angle $\varphi$ could be used to estimate not only wind speed and direction but also

space-time distributions of estimates of wind turbulence parameters. These parameters are the dissipation rate of turbulence energy $\varepsilon(h,t)$, the variance of the radial velocity $\bar{\sigma}_r^2(h,t)$, and the integral scale of longitudinal correlation of turbulent fluctuations of radial velocity $L_V(h,t)$. If the angle is $\varphi = 35.3°$, then, with an allowance made for the relation $E = (3/2)\bar{\sigma}_r^2$ (Eberhard et al., 1989), two-dimensional TKE distributions $E(h,t)$ can be assessed as well.

The method for obtaining the time series of the turbulent mixing layer height $h_{\mathrm{mix}}(t)$ from PCDL data measured by

conical scanning by a probing beam consists of the following. A probing beam is rotated around the vertical axis $z$ at the angle $\varphi$ to the horizontal with a constant angular rate and the azimuth angle $\theta$ (the angle between the projection of the beam axis on the horizontal plane and the axis $x$) varying from 0° to 360°. During the scanning, the probing volume moves at the height $h = R\sin\varphi$ along the circle of the base of the probing cone at a distance $R$ from the lidar.

After the primary processing of coherently detected echo signals of PCDL, we obtained arrays of estimates of the signal-

to-noise ratio $\mathrm{SNR}(R_k,\theta_m;n)$ and the radial velocity $V_L(R_k,\theta_m;n)$. Here, SNR is the ratio of the average heterodyne signal power to the noise power in a 50-MHz bandwidth, and the radial velocity is a projection of the wind vector onto the optical axis of the probing beam. The estimates of SNR and radial velocity $V_L$ are functions of the distance from the lidar to the center of probing volume $R_k = R_0 + k\Delta R$, azimuth angle $\theta_m = m\Delta\theta$, and the scan number $n$. Here, $k = 0, 1, 2, ..., K-1$; $\Delta R$ is the range gate length; $m = 0, 1, 2, ..., M-1$; $\Delta\theta = 360/M$ is the resolution in the azimuth angle, $n = 1, 2, 3, 4, ..., N$.

The method (Smalikho and Banakh, 2017) for determining wind turbulence parameters from the array of lidar estimates of the radial velocity obtained with conical scanning is applicable if the probability $P_b$ of a bad (false) estimate of the radial velocity is close to zero (for example, at $P_b \leq 10^{-4}$). The instrumental error $\sigma_e$ of a good estimate of the radial velocity (Frehlich and Yadlowsky, 1994; Banakh and Smalikho, 2013) and the probability $P_b$ depend on the signal-to-noise ratio





SNR , which decreases with distance. The smaller the SNR , the larger $\sigma_e$ and $P_b$ . Thus, the maximum range for the probing of wind turbulence $R_{K-1}$ is determined by the value of SNR . The distances $R_k$ correspond to heights $h_k = R_k \sin\varphi$ .

On the assumption that the wind is a stationary process (within one hour) and statistically homogeneous along the horizontal (within the circle of the base of the scanning cone), the array $V_L(R_k, \theta_m; n)$ was used to estimate the vector of the

mean wind velocity $< \mathbf{V}(h_k) > = \{<V_z>, <V_x>, <V_y>\}$ , where $V_z$ is the vertical component and $V_x$ , $V_y$ are the horizontal components of the wind vector $\mathbf{V} = \{V_z, V_x, V_y\}$ , by the sine wave fitting method (Banakh and Smalikho, 2013). Angular brackets indicate the average of an ensemble of realizations. Then, the array of random components of estimates of radial velocities is calculated as

$$V_L'(R_k, \theta_m; n) = V_L(R_k, \theta_m; n) - \mathbf{S}(\theta_m) < \mathbf{V}(h_k) >_N , \tag{1}$$

where $\mathbf{S}(\theta_m) = \{\sin\varphi, \cos\varphi\cos\theta_m, \cos\varphi\sin\theta_m\}$ is the unit vector along the optical axis of the probing beam, and

$$< f(n) >_N = \frac{1}{N} \sum_{n'=n-N/2}^{n+N/2-1} f(n+n')$$ is the average of $N$ scans. The averaged (over all azimuth angles $\theta_m$ ) variance $\bar{\sigma}_L^2$ and the azimuthal structure function $\bar{D}_L(\psi_l)$ of the fluctuations of radial velocity measured by the lidar are calculated from this array for every height $h_k$ by the following equations:

$$\bar{\sigma}_L^2 = < \frac{1}{M} \sum_{m=0}^{M-1} [V_L'(R_k, \theta_m; n)]^2 >_N , \tag{2}$$

$$\bar{D}_L(\psi_l) = < \frac{1}{M-l} \sum_{m=0}^{M-l-1} [V_L'(R_k, \theta_m + \psi_l; n) - V_L'(R_k, \theta_m; n)]^2 >_N , \tag{3}$$

where $\psi_l = l\Delta\theta$ and $l = 1, 2, 3, 4, \dots$ .

According to Smalikho and Banakh, 2017, the turbulence energy dissipation rate $\varepsilon$ is determined by the azimuthal structure function $\bar{D}_L(\psi_l)$ , which is calculated from the lidar data measured within the inertial range of turbulence by the following equation:

$$\varepsilon = \left[ \frac{\bar{D}_L(\psi_l) - \bar{D}_L(\psi_1)}{A(l\Delta y_k) - A(\Delta y_k)} \right]^{3/2} , \tag{4}$$

where $A(y)$ is the theoretically calculated function, the equation for which can be found in (Smalikho and Banakh, 2017), and $l \geq 2$ . Then, the variance of radial velocity fluctuations $\bar{\sigma}_r^2$ averaged over all azimuth angles $\theta_m$ is estimated as (Smalikho and Banakh, 2017)





$$\bar{\sigma}_r^2 = \bar{\sigma}_L^2 - \bar{D}_L(\psi_1)/2 + \varepsilon^{2/3}[F(\Delta y_k) + A(\Delta y_k)/2] \, . \tag{5}$$

The function $F(y)$ in Eq. (5) is defined in Smalikho and Banakh, 2017.

Equations (4) and (5) are used to obtain estimates of $\bar{\sigma}_r^2$ and $\varepsilon$ at different heights $h_k$ and at different instants $t_{n'} = n'\Delta t$, where $n' = 0,1,2,...,N'$, $\Delta t$ is defined by the duration of the scan $\Delta t \approx T_{\text{scan}}$, and $N'$ depends on the duration of

measurements. For 24-hour measurements, $N'$ can be found from the equation $N'\Delta t = 24$ h. The mixing layer height $h_{\text{mix}}$ for every instant $t_{n'}$ is determined from the vertical profiles of $\bar{\sigma}_r^2$ or $\varepsilon$ obtained for this instant at the height, where $\bar{\sigma}_r^2$ or $\varepsilon$ decrease with height $h_k$ down to the corresponding minimum threshold values $\bar{\sigma}_r^2(h_{\text{mix}}) = Thr_\sigma$ or $\varepsilon(h_{\text{mix}}) = Thr_\varepsilon$, at which the turbulence intensity is already insufficient for efficient mixing of air.

The algorithm for the evaluation of the mixing layer height is based on the serial search of values of $\bar{\sigma}_r^2(h_k)$ or $\varepsilon(h_k)$ at

different heights $h_k$, starting from the minimum height $h_0$ up to the height at which the velocity variance or the dissipation rate decreases to the threshold $Thr_\sigma$ or $Thr_\varepsilon$, respectively. When assessing the time series of the mixing layer height $h_{\text{mix}}(t_{n'})$ from the height–temporal distributions $\varepsilon(h_k, t_{n'})$, we use $Thr_\varepsilon = 10^{-4}$ m$^2$/s$^3$, which corresponds to the lower boundary of moderate turbulence. With weak turbulence $\varepsilon < 10^{-4}$ m$^2$/s$^3$, the turbulent mixing of air may be considered to be insignificant. The same threshold was used in Vakkari et al., 2015.

**3 Evaluation of the turbulent mixing layer height during forest fires in Siberia in 2019**

To study the atmospheric boundary layer in the air with intense smoke due to forest fires in Siberia in 2019, we conducted a lidar experiment on the measurement of wind turbulence parameters and determination of diurnal variations of the mixing layer height. Continuous measurements by the Stream Line lidar (Halo Photonics, Brockamin, Worcester, United Kingdom) were carried out from July 20 to 29 of 2019 in the territory of the Basic Experimental Observatory (BEO) of the Institute of

Atmospheric Optics SB RAS in Tomsk suburbs (56.481430 N, 85.099624 E). During the experiment, the probing beam was focused to a distance of 500 m. Conical scanning by the probing beam around the vertical axis at the alternating elevation angles 35.3° and 60° was used. For the accumulation of raw lidar data, $N_a = 7500$ and $N_a = 3000$ laser shots were used. The pulse repetition frequency was $f_p = 15$ kHz. Thus, the duration of the measurements of an array of radial velocities $V_L(R_k, \theta_m; n)$ for each azimuth angle $\theta_m$ was, respectively, $\delta t = N_a/f_p = 0.5$ and 0.2 s. The time for one scan was $T_{\text{scan}} = 60$

s. The azimuth resolution was $\Delta\theta = 360°/M = 3°$, where $M = T_{\text{scan}}/\delta t = 120$ is the number of rays per scan at $N_a = 7500$, and $\Delta\theta = 1.2°$, $M = 300$ at $N_a = 3000$. The range gate length was $\Delta R = 18$ m.

During the experiment, the optical characteristics of the atmosphere varied considerably. Most of the time, the aerosol backscatter coefficient far exceeded the background level owing to the smog from the forest fires. The signal-to-noise ratio



SNR was abnormally high for lidars such as Stream Line (pulse energy about 10 $\mu$ J) and achieved 0 dB in the cloudless atmosphere at a height of 500 m when scanning at an elevation angle of 60°. Under these conditions, with the method of filtered sine wave fitting (FSWF) (Smalikho, 2003), we succeeded in retrieving the vertical profiles of wind speed and direction up to a height of ~3 km. In the last two days of the experiment, the smog disappeared, and the atmosphere became

so clear that the echo from distances exceeding 500 m was very weak: SNR < -15 dB. Estimates of wind turbulence parameters from the data obtained at this SNR have a relative error exceeding 30%. In some time intervals, the lidar measurements were carried out under conditions of dense fog or low cloudiness, which were serious obstacles to obtaining information about wind and turbulence in the entire atmospheric boundary layer.

Each of the obtained arrays of estimates of the signal-to-noise ratio $\mathrm{SNR}(R_k, \theta_m; n)$ and the radial velocity $V_L(R_k, \theta_m; n)$

contained two sub-arrays, $\mathrm{SNR}_i(R_k, \theta_m; n)$ and $V_{Li}(R_k, \theta_m; n)$, where the subscript $i = 1$ corresponds to measurements at the elevation angle $\varphi = \varphi_1 = 35.3°$ (odd scan numbers $n$), and $i = 2$ corresponds to measurements at $\varphi = \varphi_2 = 60°$ (even $n$). The elevation angle $\varphi$ was alternated for $\Delta \tau \approx 1.5$ s. The height–temporal distributions of the absolute value of the speed $U_i(h_{ki}, t_{n'})$ and direction angle $\theta_{Vi}(h_{ki}, t_{n'})$ of the horizontal wind and the vertical wind velocity $W_i(h_{ki}, t_{n'})$, where $h_{ki} = R_k \sin \varphi_i$, $t_{n'} = t_{0i} + n'2(T_{\mathrm{scan}} + \Delta \tau)$, $n' = 0,1,2,...$, were calculated from the arrays $V_{Li}(R_k, \theta_m; n)$ through the methods of

direct and filtered sine wave fitting (Banakh and Smalikho, 2013). The height–temporal distributions of the signal-to-noise ratio $\mathrm{SNR}_i(h_{ki}, t_{n'})$ for every $n$-th scan were found as a result of averaging $\mathrm{SNR}_i(R_k, \theta_m; n)$ over all the azimuth angles $\theta_m$.

## 4 Results of the experiment

Figure 1 shows the height–temporal distributions $\mathrm{SNR}_i(h_{ki}, t_{n'})$, $U_i(h_{ki}, t_{n'})$, $\theta_{Vi}(h_{ki}, t_{n'})$, and $W_i(h_{ki}, t_{n'})$ obtained from measurements on July 21 of 2019. On this day, there were no clouds, and the smog was observed from 00:00 to 24:00 Local

Time up to a height of at least 2 km. Owing to the smog, the signal-to-noise ratio was high, and at an elevation angle of 60°, it exceeded -10 dB for the entire day in the 1-km atmospheric layer adjacent to the ground. Most of the time, in the layer above 500 m, $\mathrm{SNR}_1(h) < \mathrm{SNR}_2(h)$ at the same height $h$ since the echo signal travels a longer distance at smaller elevation angles. The analysis of wind data for this day shows that for the 30-min moving average of lidar estimates of wind velocity vector components, there are practically no differences between $U_1(h,t)$ and $U_2(h,t)$ or between $\theta_{V1}(h,t)$ and $\theta_{V2}(h,t)$ up

to a height of 1200 m.

From the obtained arrays of lidar estimates of the radial velocity $V_{L1}(R_k, \theta_m; n)$ and $V_{L2}(R_k, \theta_m; n)$, the height–temporal distributions of the turbulence energy dissipation rate $\varepsilon_i(h_{ki}, t_{n'})$ and the variance of radial velocity $\bar{\sigma}_{ri}^2(h_{ki}, t_{n'})$ up to heights of 1000 m ($i = 1$, elevation angle $\varphi = 35.3°$) and 1500 m ($i = 2$, $\varphi = 60°$) were calculated by Eqs. (1)-(5). In the calculations with Eqs. (2) and (3), we took $N = 15$. For scanning at two angles at $T_{\mathrm{scan}} = 60$ s, this corresponds to the





approximately 30-min average of measured data. The calculated results are shown in Figs. 2 and 3. The black color in these and subsequent figures represents a lack of data because of their low quality owing to an insufficiently high signal-to-noise ratio or because the parameter under consideration is smaller than the lower boundary shown in the color scale.

A comparison of the data in Figs. 2(a) and 2(b) for the lower 1-km layer of the atmosphere demonstrates the closeness of the estimates $\varepsilon_1(h,t)$ and $\varepsilon_2(h,t)$ obtained from measurements at different elevation angles, which is in agreement with the results of Banakh and Smalikho, 2019 and confirms the assumption of horizontal homogeneity of the turbulent wind field. The difference in the estimates of the radial velocity $\bar{\sigma}_{r1}^2(h,t)$ and $\bar{\sigma}_{r2}^2(h,t)$ measured at different elevation angles is more significant than that for the dissipation rate and is caused by the anisotropy of wind turbulence (Banakh and Smalikho, 2019).

Figure 4 shows the diurnal series of the turbulence energy dissipation rate $\varepsilon(t)$ and turbulence kinetic energy $E(t)$ at different heights of the atmospheric boundary layer. Here, we used the data in Figs. 2(b) and 3(b) with an allowance for the relation $E = (3/2)\bar{\sigma}_r^2$ for an elevation angle of 35.3° (Eberhard et al., 1989). One can see that at a height of 60 m, the dissipation rate for the whole day exceeded $10^{-4}$ m²/s³, whereas, at a height of 900 m, this occurred only in relatively short time intervals between 13:00 and 17:00 Local Time. At approximately 12:45, the kinetic energy $E$ was approximately the same at different heights in the layer from 60 to 900 m, whereas the turbulence energy dissipation rate $\varepsilon$ decreased with height.

Figure 5 exemplifies the vertical profiles $\varepsilon(h_k)$ for the case in which the dissipation rate nearly coincides with the threshold $Thr_\varepsilon = 10^{-4}$ m²/s³ in the lower (60 m) and upper (1 km) levels. From the data shown in Fig. 2b, in the measurements taken at an elevation angle of 35.3°, the dissipation rate in the upper level exceeds this threshold in some periods, in contrast to the measurements at an elevation angle of 60° (Fig. 2a).

The mixing layer height $h_{\mathrm{mix}}$ was determined from the obtained height–temporal distributions of $\varepsilon_i(h_k,t_{n'})$ and $\bar{\sigma}_{ri}^2(h_k,t_{n'})$ for every instant $t_{n'}$ with the use of the relations $\bar{\sigma}_r^2(h_{\mathrm{mix}}) = Thr_\sigma = 0.1$ m²/s² and $\varepsilon(h_{\mathrm{mix}}) = Thr_\varepsilon = 10^{-4}$ m²/s³. The maximum height of the estimation of the temporal MLH series was 1 km for the measurements at an elevation angle of 35.3° and 1.5 km for the measurements at an angle of 60°. The minimum height was $h_0 = 60$ m. If the estimates of $\bar{\sigma}_r^2(h_0,t_{n'})$ or $\varepsilon(h_0,t_{n'})$ at a height of 60 m were smaller than the corresponding threshold, then we took $h_{\mathrm{mix}} = h_0 = 60$ m. If the estimates of $\bar{\sigma}_r^2(h_0,t_{n'})$ or $\varepsilon(h_0,t_{n'})$ at the maximum height exceeded the threshold, we took $h_{\mathrm{mix}}$ to be equal to the maximum height of restoration of the vertical profiles of turbulence parameters, which could be smaller than 1 km at $\varphi = 35.3°$ and smaller than 1.5 km at $\varphi = 60°$ because of the low quality of measured data.

Figure 6 shows the diurnal time series of $h_{\mathrm{mix}}(t_{n'})$ obtained from the height–temporal distributions of the dissipation rate and the variance of radial velocity shown in Figs. 2 and 3. One can see that for the diurnal series of the turbulent mixing layer height retrieved from measurements of the dissipation rate at elevation angles of 35.3° and 60°, we have, with rare





exceptions, rather close results. The temporal series $h_{\mathrm{mix}}(t_{n'})$ calculated from the variances differ more widely as a result of turbulence anisotropy (Banakh and Smalikho, 2019).

Since the temporal MLH series found from estimates of the dissipation rate at different elevation angles differ insignificantly, for other days of the experiment, we calculated $h_{\mathrm{mix}}(t_{n'})$ from the estimates of the dissipation rate obtained

by scanning at an elevation angle 60°. The signal-to-noise ratio SNR at heights above 500 m is markedly higher at an elevation angle of 60° than at $\varphi = 35.3°$ (Figs. 1a and 1e). Thus, measurements at 60° provide an estimation of turbulence intensity at higher levels.

Figure 7 shows the height–temporal distributions of the signal-to-noise ratio and the turbulence energy dissipation rate obtained from lidar data for July 20 of 2019. On this day, in contrast to July 21 of 2019, the lidar measurements were

conducted under conditions of a partly cloudy atmosphere. As a probing pulse enters a cloud, the signal-to-noise ratio sharply increases. However, SNR then decreases rapidly while the pulse propagates in the cloud, and the lidar begins to measure only noise. Figure 7a shows that the signal-to-noise ratio varied just in this way on the night of July 20. We can see that there were clouds at heights above 1 km until 03:00 in the morning. After 06:00 in the morning, low clouds (fog) were observed. Then, in the process of surface heating by solar radiation and the development of convection, the clouds rose and

disappeared completely at 10:00. In the period from 13:00 to approximately 19:00 LT, clouds sometimes appeared at heights above 1.5 km. After 19:00, the signal-to-noise ratio increased in the 700-m layer adjacent to the Earth's surface because of the intense smog transported by the wind to the region of the experiment. The presence of clouds explains the large gap in data (black color) in Fig. 7b.

The MLH time series $h_{\mathrm{mix}}(t_{n'})$ calculated from the distribution of the dissipation rate in height and time in Fig. 7b is

shown by the gray curve in Fig. 7a and red curve in Fig. 7b. One can see that MLH reaches its maximum of about 1800 m at approximately 15:00.

The height–temporal distributions of the signal-to-noise ratio and the turbulence energy dissipation rate obtained from lidar measurements on July 23 of 2019 are shown in Fig.8. On this day, fog was observed from 05:00 to 07:00 in the morning, and the rest of the time, the atmosphere was partly cloudy at heights above 800 m. The signal-to-noise ratio in the

layer adjacent to the surface was markedly lower than that on the previous days. The temporal series of MLH $h_{\mathrm{mix}}(t_{n'})$ in Fig. 8 is shown by the gray and red curves. It can be seen that $h_{\mathrm{mix}}(t_{n'})$ differs from the minimum height of restoration of vertical profiles of the dissipation rate $h_0 = 60$ m in the period from 09:00 to 20:00. For the rest of the time, turbulence was very weak, with the dissipation rate $\varepsilon < Thr_\varepsilon = 10^{-4}$ m$^2$/s$^3$. At approximately 16:00, MLH was at its maximum and reached 1400 m.

The height and time distributions of the signal-to-noise ratio and the turbulence energy dissipation rate obtained from lidar measurements on July 25 of 2019 are shown in Fig. 9. According to Fig. 9a, on this day, between 08:20 and 13:30, it was cloudy with the rise of the cloud base because insolation increased and convection developed during this time. In the absence





of clouds, at the heights in the 1-km layer adjacent to the Earth's surface and at heights below clouds, the signal-to-noise ratio was no lower than -10 dB, and the accuracy of the estimation of wind turbulence parameters was high. Since turbulence was weak from 00:00 to 07:00 and from 20:40 to 23:20 (the dissipation rate was $\varepsilon < Thr_\varepsilon = 10^{-4}$ m²/s³), MLH estimates for these periods are equal to the minimum height of restoration of vertical profiles of turbulence (60 m). It can be seen in Fig.9b

that between 11:00 and 16:00, $h_{mix}$ varied mostly from 1000 to 1100 m and was at the maximum value of 1200 m at 14:00.

The diurnal time series of MLH obtained from lidar measurements on July 20, 21, 23, and 25 of 2019 are summarized in Fig. 10. It follows from Fig. 10 that between 13:00 and 17:00 in the daytime, the mixing layer height varied widely: from 550 to 1800 m. From 00:00 to 07:00 in the morning and 21:00 to 24:00 in the evening, when the temperature stratification, according to data of sonic anemometers employed in this experiment, was stable, the MLH estimate was taken to be equal to

the smallest height $h_0 = 60$ m of restoration of vertical profiles of the turbulence energy dissipation rate. As a rule, the estimate of the dissipation rate in these time intervals was smaller than $10^{-4}$ m²/s³. The choice of the minimum height $h_0 = 60$ m in the measurements at the elevation angle $\varphi = 60°$ is explained by the fact that the minimum range of the pulsed coherent Doppler lidar should be no smaller than the two lengths of the probing volume (Smalikho et al., 2015). In our measurements, the Stream Line lidar formed a probing volume with a length of 30 m.

One can see from Fig. 10 that on July 20 and 25, the time series of MLH are practically identical during the period from 8:00 to almost 12:00. Recall that in both cases, the increase in $h_{mix}$ was accompanied by the rise of the cloud base due to convection (see Figs. 7a and 9a). This confirms the correctness of the MLH time series assessment based on height–temporal distributions of the turbulence energy dissipation rate.

**5 Error of MLH estimation**

The accuracy of MLH estimation from the PCDL data obtained with the use of conical scanning is determined by the error of estimation of the turbulence energy dissipation rate in the relatively thin atmospheric layer centered at the height $h_{mix}$. To determine this error, we calculated not only height–temporal distributions of the dissipation rate $\varepsilon(h_k, t_{n'})$ but also instrumental errors of lidar estimation of the radial velocity $\sigma_e(h_k, t_{n'})$ by Eq. (23) from Smalikho and Banakh, 2017. Then, the relative errors of lidar estimation of the turbulence energy dissipation rate $E_\varepsilon(h_k, t_{n'})$ were calculated with the use of the

distributions of the dissipation rate $\varepsilon(h_k, t_{n'})$ and the instrumental error $\sigma_e(h_k, t_{n'})$ by Eqs. (6)-(11) from Banakh et al., 2017, and the time series of this error $E_\varepsilon(h_{mix}(t_{n'}))$ at heights $h_{mix}(t_{n'})$ were determined.

Figure 11 shows the behavior of the relative error $E_\varepsilon(h_{mix}(t_{n'}))$ for the diurnal time series of MLH shown in Fig. 10. It follows from Fig. 11a that the relative errors $E_\varepsilon(h_{mix}(t_{n'}))$ exceed 30% for measurements on July 20 in the period between 00:00 and 06:00. This is explained by the relatively large instrumental error of estimation of the radial velocity due to the

low signal-to-noise ratio ( SNR < -15 dB, see Fig. 7a). On the same day, in the period between 11:00 and 18:00, the signal-





to-noise ratio at heights above 1 km did not exceed -10 dB (Fig. 7a) and sometimes decreased to the lowest threshold SNR = -16 dB, at which the probability of a bad (false) estimate of the radial velocity $P_b$ can still be considered close to zero. As a result, the instrumental error of estimation of the radial velocity $\sigma_e$ is much larger than that at heights below 1 km. As the mixing layer height increases, SNR decreases, and the error $\sigma_e$ increases too (and vice versa for a decrease in $h_{\text{mix}}$ ). This

explains the initial increase in the relative error of estimation of the dissipation rate $E_\varepsilon$ and its subsequent decrease in the considered period between 11:00 and 18:00. In the other periods, as can be seen from Fig. 11a, the error $E_\varepsilon$ is about 10%, which provides high accuracy of the determination of the mixing layer height.

  On July 21, the signal-to-noise ratio in the 1-km layer adjacent to the Earth's surface was very high most of the time (see Fig. 1a). Correspondingly, the instrumental error of estimation of the radial velocity $\sigma_e$ within the mixing layer did not

exceed 0.1 m/s. Therefore, according to the data in Fig. 11b, the error of estimation of the dissipation rate $E_\varepsilon (h_{\text{mix}})$ was low (mostly about 10%) and did not exceed 18%.

  On July 23, the signal-to-noise ratio was low. Within the mixing layer, SNR ~ -10 dB at a height of 100 m and SNR ~ -15 dB at a height of 1 km (see Fig. 8a). As a result, the relative error $E_\varepsilon (h_{\text{mix}})$ varied widely from 10% to 30% (mostly larger than 15%), as is indicated by Fig.11c.

In the period between 08:00 and 20:40 on July 25, the dissipation rate was determined with very high accuracy. The relative error $E_\varepsilon (h_{\text{mix}})$ did not exceed 15% (mostly 9%), which is explained by the very high signal-to-noise ratio (see Fig. 9a). Despite the fact that the SNR was also rather high for the rest of the time, the accuracy of the dissipation rate estimation in the periods from 00:00 to 07:00 and from 21:00 to 23:00 on July 25 was low, and the relative error $E_\varepsilon (h_{\text{mix}})$ exceeded 30%. The reason for this is that the dissipation rate at the height $h_{\text{mix}} = h_0 = 60$ m during this time, according to

Fig. 9b, was smaller by approximately an order of magnitude than the threshold value $Thr_\varepsilon = 10^{-4}$ m$^2$/s$^3$.

  With the use of the algorithm in Smalikho and Banakh, 2013, we conducted a series of closed numerical experiments on the retrieval of vertical profiles of the turbulence energy dissipation rate $\varepsilon(h_k)$ from simulated lidar data. The simulation was performed for different values of the signal-to-noise ratio SNR and the vertical gradient of the dissipation rate $\gamma = -d\varepsilon / dh > 0$ at the height $h$, where the dissipation rate was set equal to $\varepsilon = 10^{-4}$ m$^2$/s$^3$. The turbulent mixing layer

height was estimated from the profiles of $\varepsilon(h_k)$ obtained in the numerical experiments with the use of the threshold $Thr_\varepsilon = 10^{-4}$ m$^2$/s$^3$. The obtained estimates $\hat{h}_{\text{mix}}$ were compared with the preset values $h_{\text{mix}}$. The analysis of results of the numerical experiments shows that the accuracy of estimates $\hat{h}_{\text{mix}}$ depends significantly not only on SNR but also on the vertical gradient of the dissipation rate $\gamma$. The smaller the value of $\gamma$ (the slower the decrease in the dissipation rate with height), the larger the error $\sigma_h = \sqrt{< (\hat{h}_{\text{mix}} - h_{\text{mix}})^2 >}$ .



Calculation of the experimental error $\sigma_h$ with the algorithm in Smalikho and Banakh, 2013 would require very computationally expensive simulation. The experimental error of MLH estimation was determined in another way. To this end, on the assumption that $E_\varepsilon(h_k) < 30\%$, the random estimate of the dissipation rate $\hat{\varepsilon}(h_k)$ at the height $h_k$ was taken as

$$\hat{\varepsilon}(h_k) = \varepsilon(h_k)\left[1 + \frac{E_\varepsilon(h_k)}{100\%}\xi(h_k)\right],$$ (6)

where $\varepsilon(h_k)$ and $E_\varepsilon(h_k)$ are respectively the estimate of the dissipation rate and its relative error obtained from the data of the lidar experiment, and $\xi(h_k)$ is a computer-generated random number from the normal distribution of the probability density function with zero mean $<\xi> = 0$ and unit variance $<\xi^2> = 1$. To construct the vertical profile $\hat{\varepsilon}(h_k)$, random numbers $\xi(h_k)$ for different heights $h_k$ were generated in accordance with the correlation function $C_\xi(l\Delta h) = <\xi(h_{\text{mix}} + l\Delta h/2)\xi(h_{\text{mix}} - l\Delta h/2)>$, where $h_{\text{mix}}$ is the turbulent mixing layer height determined from the
experiment, and $\Delta h = 15.6$ m is a step in height at $\Delta R = 18$ m and $\varphi = 60°$. The correlation function $C_\xi(l\Delta h)$ was found from the numerical experiments. For the error $E_\varepsilon(h_k) < 30\%$ and $Thr_\varepsilon = 10^{-4}$ m²/s³, the correlation function $C_\xi(l\Delta h)$ weakly depends on SNR and $h_{\text{mix}}$, which considerably simplifies the procedure of numerical simulation of $\hat{\varepsilon}(h_k)$ by Eq. (6).

The accuracy of retrieval of the temporal profiles of the turbulent mixing layer height was analyzed for the measurements
on July 20 and 21 of 2019, when the dissipation rate $\varepsilon(h_0, t_{n'})$ at the initial height $h_0 = 60$ m exceeded the threshold value $Thr_\varepsilon = 10^{-4}$ m²/s³ all day. Figures 12 and 13 show the diurnal profiles of MLH $h_{\text{mix}}(t_{n'})$ and the absolute $\sigma_h(t_{n'})$ and relative $E_h(t_{n'})$ errors of its estimation, as determined from the data measured on these days. The procedure for calculating the absolute and relative errors of MLH is the following.

According to the algorithm of MLH estimation, the differences $h_{\text{mix}} - h_0$ (see, for example, Fig. 10) are multiples of the
step in height $\Delta h$. Upon vertical interpolation of the experimental errors $\varepsilon(h_k)$ and passage from the discrete vertical profile $\varepsilon(h_k)$ to the continuous one $\varepsilon(h)$, we determined $h_{\text{mix}}$ from the equality $\varepsilon(h_{\text{mix}}) = Thr_\varepsilon = 10^{-4}$ m²/s³, where $h_{\text{mix}}$ is the smallest height satisfying this equality. The results of the restoration of the diurnal time series $h_{\text{mix}}(t_{n'})$ with the use of the interpolated profiles $\varepsilon(h_k)$ are depicted in Figs. 12a and 13a. A comparison of the blue and red curves in Fig. 10 with the curves in Figs. 12a and 13a demonstrates that the latter curves are more smoothed. In calculations of the errors, the values of
$h_{\text{mix}}$ found in this way were taken as the true mixing layer heights. From random realizations of $\hat{\varepsilon}(h_k)$ generated by the computer with Eq. (6), we obtained MLH estimates $\hat{h}_{\text{mix}}$ by the algorithm described in Section 2 and then calculated the absolute error $\sigma_h$ and the relative error $E_h = (\sigma_h/h_{\text{mix}})\times100\%$ of lidar estimation of the turbulent mixing layer height. The



averaging was performed over $10^5$ independent realizations of $\xi(h_k)$ for every pair of vertical profiles $\varepsilon(h_k, t_{n'})$ and $E_\varepsilon(h_k, t_{n'})$ obtained in the atmospheric experiment at different times $t_{n'}$.

One can see from Fig. 12b that in the periods from 00:00 to 16:30 and from 18:00 to 24:00 LT on July 21 of 2019, the error $\sigma_h(t_{n'})$ varied mostly from 3 to 10 m, rarely exceeding 20 m. However, in the period between 16:30 and 18:00, the

error of MLH estimation increased sharply, achieving $\sigma_h \sim 70$ m. This was caused by the very slow decrease in the turbulence energy dissipation rate with height in this period (the gradient $\gamma$ at the mixing layer height was small). Nevertheless, as can be seen from Fig. 12c, from 16:30 to 18:00, the relative error $E_h(t_{n'})$ was mostly below 15%. With the rare exception, the relative error $E_h(t_{n'})$ remained within 15% all day on July 21 as well. At midday, when the mixing layer height was at the maximum, the relative error $E_h$ decreased to 1%.

As can be seen from the data in Fig. 7a, during the night period on July 20 of 2019, SNR in the mixing layer varied from -17 dB to -14 dB. As a result, the relative error of estimation of the dissipation rate $E_\varepsilon(h_{mix}(t_{n'}))$ at the mixing layer height, according to the data in Fig. 11a, often exceeds 30% in the period between 00:00 and 06:00. This does not allow us to obtain estimates of MLH for this time with an acceptable (less than 20%) relative error, as can be seen from Fig. 13c. The calculated errors $\sigma_h(t_{n'})$ and $E_h(t_{n'})$ shown in Fig. 13 for the period between 00:00 and 06:00 may be of limited accuracy

because the condition of the applicability of Eq. (6) $E_\varepsilon(h_{mix}(t_{n'})) < 30\%$ was not always fulfilled in this period. A considerable (up to 30%) increase in the error $E_\varepsilon(h_{mix}(t_{n'}))$ in the period 13:00-16:00 LT, as follows from Fig. 11a, causes an increase in the absolute error of the MLH estimation $\sigma_h(t_{n'})$ up to ~100 m in this period (Fig. 13b). However, since the mixing layer height $h_{mix}(t_{n'})$ during this time reached 1600-1800 m, the relative error $E_h(t_{n'})$ was less than 10%. In general, according to the data in Fig.13c, the relative error of estimation of the turbulent mixing layer height $E_h(t_{n'})$ was quite

acceptable in the period from 06:00 to 24:00 on July 20 of 2019 and did not exceed 10%, with rare exceptions.

## 6 Comparison with the Richardson number

One of the parameters characterizing the static stability of the atmospheric boundary layer is the gradient Richardson number,

$$\text{Ri} = N^2 \left(\frac{\partial U}{\partial h}\right)^{-2},$$    (7)

where

$$N^2 = \frac{g}{T_p}\frac{\partial T_p}{\partial h},$$    (8)





$T_p(h,t) = T(h,t) + \gamma_a h$ is the potential temperature, $T(h,t)$ is the temperature of the air, $\gamma_a = 0.0098$ deg/m is the dry-adiabatic gradient, and $\partial T_p(h,t)/\partial h = \partial T(h,t)/\partial h + \gamma_a$. The gradient Richardson number can be used to estimate the turbulent mixing layer height (Helmis et al., 2012; Petenko et al., 2019; Gibert et al., 2011).

For additional proof of the suitability of the method for estimating the turbulent mixing layer height from lidar data obtained with the use of conical scanning, we conducted a lidar experiment with concurrent measurement of the temperature. The experiment was conducted from April 8 to May 6 of 2020. The temperature was measured by the MTP-5 microwave temperature profiler (Atmospheric Technology, Dolgoprudnyi, Moscow, Russia). The temperature profiler and the wind lidar Stream Line were installed on the roof of the Institute of Atmospheric Optics (IAO) building in Tomsk (56.475504 N, 85.048225 E ) four kilometers from the Basic Experimental Observatory. The profiler provided measurements of the vertical temperature profiles every 5 min, with a resolution of 25 m for heights from 0 to 100 m and a resolution of 50 m for heights from 100 to 1000 m with respect to the height of its installation. As a result, we obtained the height–temporal distributions of the air temperature $T(h,t)$. In calculating the derivative $\partial T_p(h,t)/\partial h$ and the parameter $N^2$ (8), we used the temperature measurement data averaged over a 10-min period. Then, the Richardson number Ri was calculated by Eq. (7). The mean horizontal wind velocity $U$ and its derivative $\partial U/\partial h$ in Eq. (7) were assessed from the lidar data averaged, as with the temperature, over a 10-min period (10 scans). The parameters and geometry of the lidar measurements were the same as those in July 2019. The scanning was carried out at an elevation angle of 60°.

During this experiment, the signal-to-noise ratio did not allow us to obtain estimates of the wind velocity with an acceptable error at heights above 1 km. The relative error of estimation of the turbulence energy dissipation rate was also large and could exceed 30% starting from heights of 400-600 m. Nevertheless, the height–temporal distributions of the dissipation rate obtained in the experiment allowed us to monitor the turbulent mixing layer height by the threshold $Thr_\varepsilon = 10^{-4}$ m²/s³ in many cases.

As an example, Fig. 14 shows the height–temporal distributions of the turbulence energy dissipation rate and the radial velocity assessed from the data of lidar measurements on May 1 of 2020. On this day, the error $E_\varepsilon$ was smaller than 30% at night between approximately 02:00 and 04:00 and between 20:00 and 22:00 within heights up to 600 and 400 m, respectively. From 05:00 to 19:00, the relative error remained below 30% at heights up to 600 m. At heights of 600-750 m, the relative error of estimation of the dissipation rate sometimes increased up to 40-50% in the period between 13:00 and 17:00. In the periods between 00:00 and 02:00, and 22:00 and 24:00, the error $E_\varepsilon$ was unacceptably large, exceeding 50%. The curves with orange (a) and black (b) squares in Fig. 14 show the diurnal time series of the turbulent mixing layer height, as estimated from the threshold values for the turbulence energy dissipation rate (orange) and for the variance of radial velocity (black).

Figure 15 shows the height–temporal distribution of the Richardson number calculated from measurements of the wind velocity and temperature for May 1. According to the classification of the atmospheric turbulent regimes based on the





Richardson number (Baumert and Peters, 2009; Grachev et al., 2013), the small-scale turbulence becomes weak at gradient Richardson numbers more than 0.5. Therefore, it is natural to believe that the turbulent mixing occurs at the time and heights at which the Richardson number Ri < 0.5. Thus, the minimum height, above which the Richardson number exceeds 0.5, can be taken as the height of the turbulent mixing layer at the chosen time. Zones in which the Richardson number Ri < 0.5 are

shown in black in the figure. One can see that from 02:00 to 04:00 and from 09:00 to 19:00, the Richardson number remained smaller than 0.5 up to a height of approximately 700 m, which can be taken as an estimate of the turbulent mixing layer height. From roughly 05:00 to 08:00, the turbulence was intermittent. Above the layer of static stability, where Ri > 0.5, the zone of turbulent mixing with Ri < 0.5 was formed at heights of 400-600 m. As can be seen from Fig. 14, the dissipation rate in the layer of static stability at a height of 200-400 m in the period from 05:00 to 08:00 remained higher

than the threshold of $10^{-4}$ m$^2$/s$^3$.

Red and white curves reproduce the diurnal time series of the turbulent mixing layer height, as estimated from the threshold values for the turbulence energy dissipation rate (red) and for the variance of radial velocity (white) depicted in Fig.14 as orange and black curve correspondingly. It can be seen that in the periods for which the error of estimation of the dissipation rate from lidar data was less than 30-50%, the estimates of the turbulent mixing layer height from the threshold

value of the dissipation rate $Thr_\varepsilon = 10^{-4}$ m$^2$/s$^3$ and from the criterion Ri < 0.5 are in a good agreement.

**7 Summary**

In this paper, we propose a method for estimating the turbulent mixing layer height on the basis of the height–temporal distributions of the turbulence energy dissipation rate obtained from PCDL measurement data with conical scanning by a probing beam around the vertical axis. The method was tested in experiments in which the atmospheric boundary layer was

investigated during smog conditions due to forest fires in Siberia in 2019. The experiments were carried out in the territory of the Basic Experimental Observatory of the Institute of Atmospheric Optics in Tomsk with the use of the Stream Line lidar (Halo Phtonics, Brockamin, Worcester, United Kingdom).

The optical characteristics of the atmosphere varied significantly during the experiments. Most of the time, the aerosol backscatter coefficient far exceeded the background level because of the smog. Under these conditions, the signal-to-noise

ratio SNR  was abnormally high for lidars in the class of the Stream Line lidar with a pulse energy of about 10 $\mu$ J. As a result, in the experiment, we succeeded in retrieving the vertical profiles of the wind speed and direction up to a height of ~3 km and wind turbulence parameters up to a height of 2 km.

The raw data of the lidar experiments conducted on July 20, 21, 23, and 25 of 2019 were used to find the diurnal time series of MLH under conditions of intense smog from the height–temporal distributions of the dissipation rate with the use of

the inequality $\varepsilon < Thr_\varepsilon = 10^{-4}$ m$^2$/s$^3$ as a criterion that indicates the absence of turbulent mixing. According to the results obtained, on these days, the MLH was at its maximum between 11:00 and 18:00 LT and varied from 550 to 1800 m, depending on the wind and turbulence intensity. It was shown in the experiment that the estimation of the turbulent mixing





layer height from the height–temporal distributions of the turbulence energy dissipation rate has some advantages in comparison with the estimation from the height–temporal distributions of the variance of radial velocity. Because of the anisotropy of wind turbulence, the variance of radial velocity depends significantly on the elevation angle of the scanning.

On July 20 and 25, the time series of MLH are practically identical during the period from 8:00 to almost 12:00. In both cases, MLH was increasing in this period, and its increase was accompanied by the rise of the cloud base due to convection. This confirms the correctness of the MLH time series assessment based on the height–temporal distributions of the turbulence energy dissipation rate.

The accuracy of the method for estimating the turbulent mixing layer height from the lidar data obtained with the use of conical scanning is discussed in detail in this paper. We developed a method for calculating the experimental error of estimation of the turbulent mixing layer height from the height–temporal distributions of the turbulence energy dissipation rate. The analysis of the errors calculated by this method shows that the accuracy of MLH estimation depends decisively on the error of estimation of the dissipation rate and on the vertical gradient of the dissipation rate at heights near the top of the mixing layer. In turn, the accuracy of estimation of the dissipation rate depends strongly on the lidar signal-to-noise ratio SNR. For the estimation of MLH with the acceptable relative error not exceeding 20%, SNR should be no less than -16 dB, when the relative error of lidar estimation of the dissipation rate does not exceed 30%. With the particular data obtained in the experiments, we demonstrate that the relative error of determination of the MLH time series from lidar measurements of the dissipation rate with the use of conical scanning does not exceed 10% in the period of turbulence development, from 06:00 to 22:00 LT. Most of the time in this period, it is less 5%.

To prove the suitability of the method for estimating the turbulent mixing layer height from lidar data obtained with the use of conical scanning, we conducted a lidar experiment with concurrent measurement of the temperature. From the obtained data, we calculated the height–temporal distributions of the gradient Richardson number and determined the MLH from these distributions. A comparison shows that the estimates of the turbulent mixing layer height from the dissipation rate distributions and from the Richardson number distributions are in good agreement.

## ACKNOWLEDGMENTS

The authors thank Artem Sherstobitov for helping with data processing. This study was supported by the Russian Science Foundation (Project No. 19-17-00170).

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



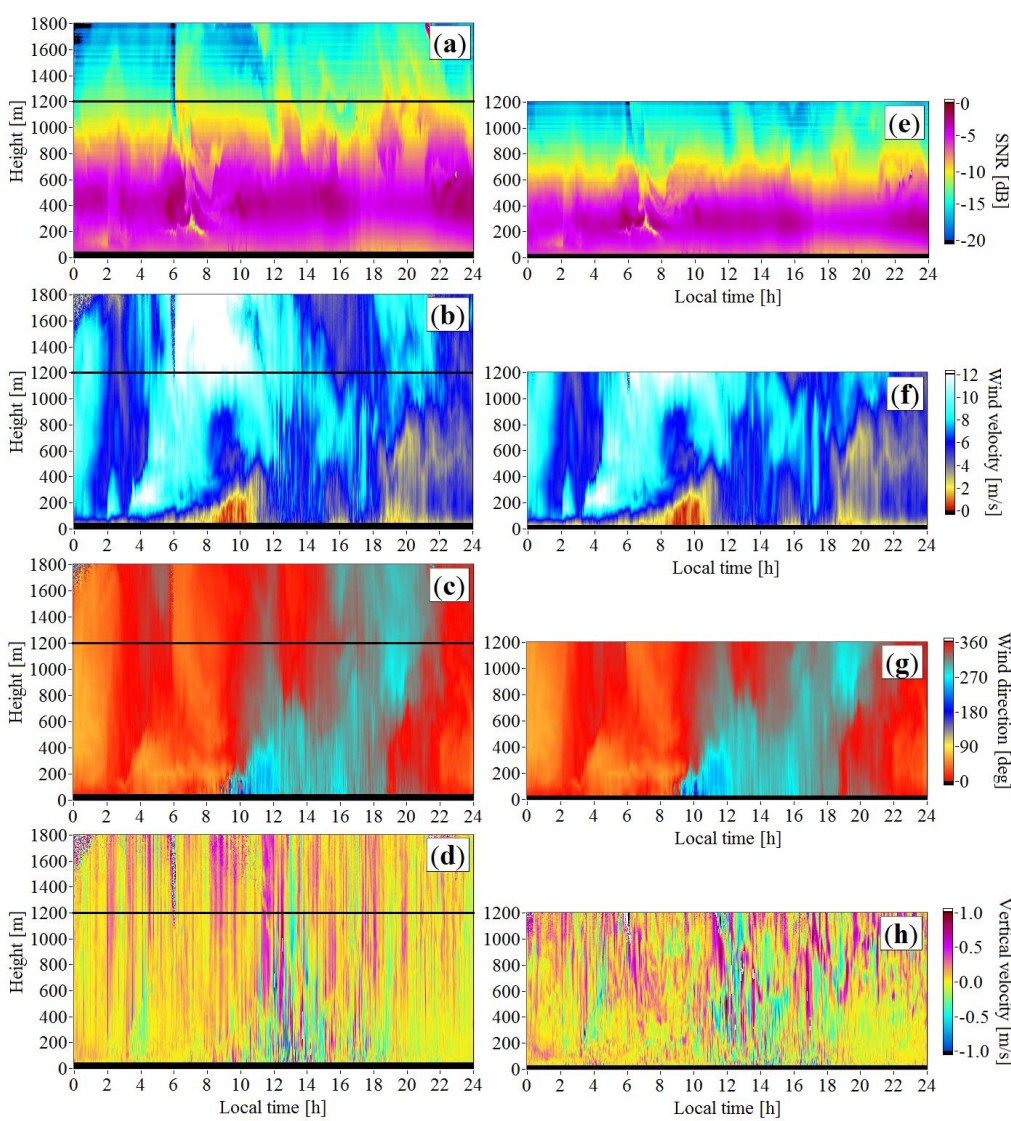

**Figure 1**: Height–temporal distributions of the signal-to-noise ratio (a, e), wind speed (b, f), wind direction angle (c, g), and vertical component of the wind vector (d, h) for elevation angles of 60° (a-d) and 35.3° (e-h) on July 21 of 2019.


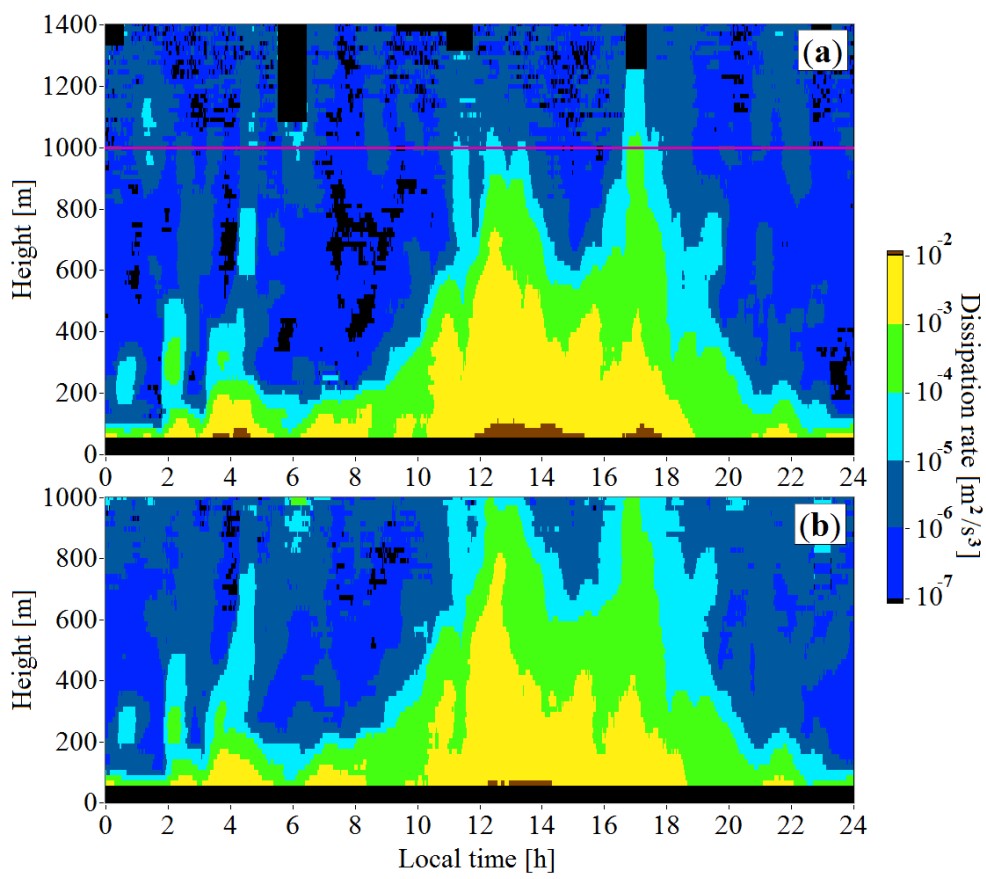

**Figure 2**: Height and time distributions of the turbulence energy dissipation rate at elevation angles of 60° (a) and 35.3° (b). Measurements were taken on July 21 of 2019.



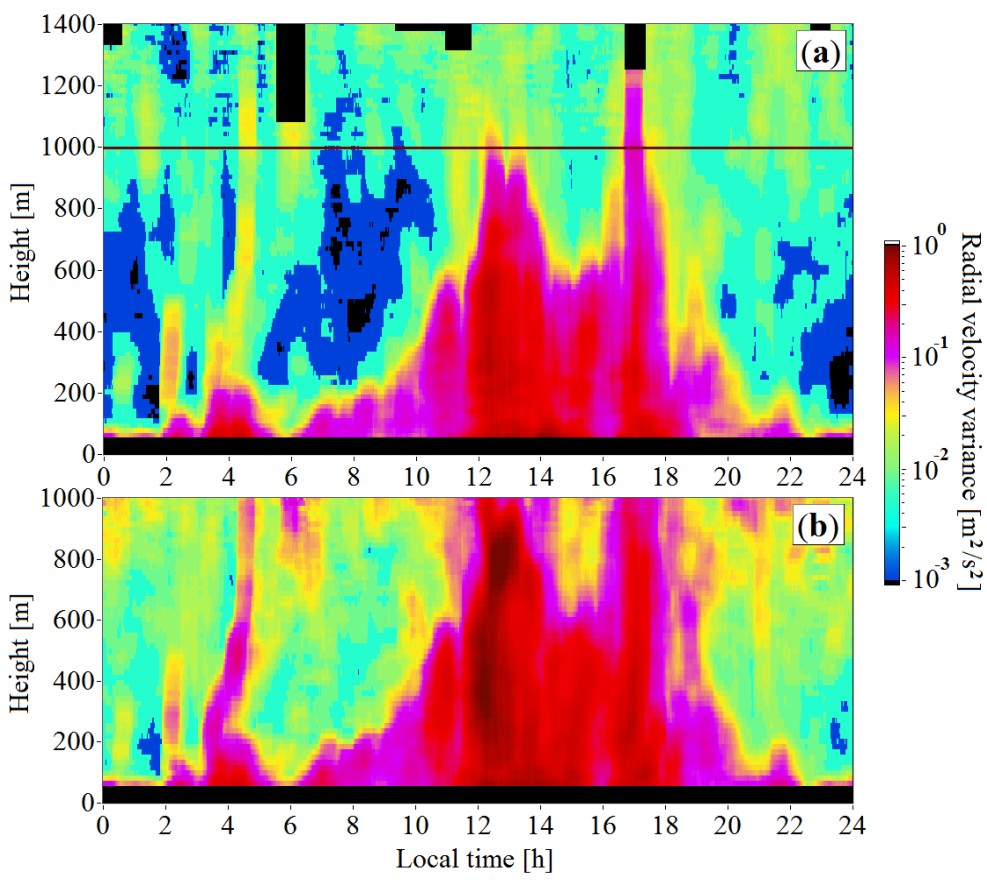

**Figure 3**: Height and time distributions of the variance of radial velocity at elevation angles of 60° (a) and 35.3° (b). Measurements were taken on July 21 of 2019.

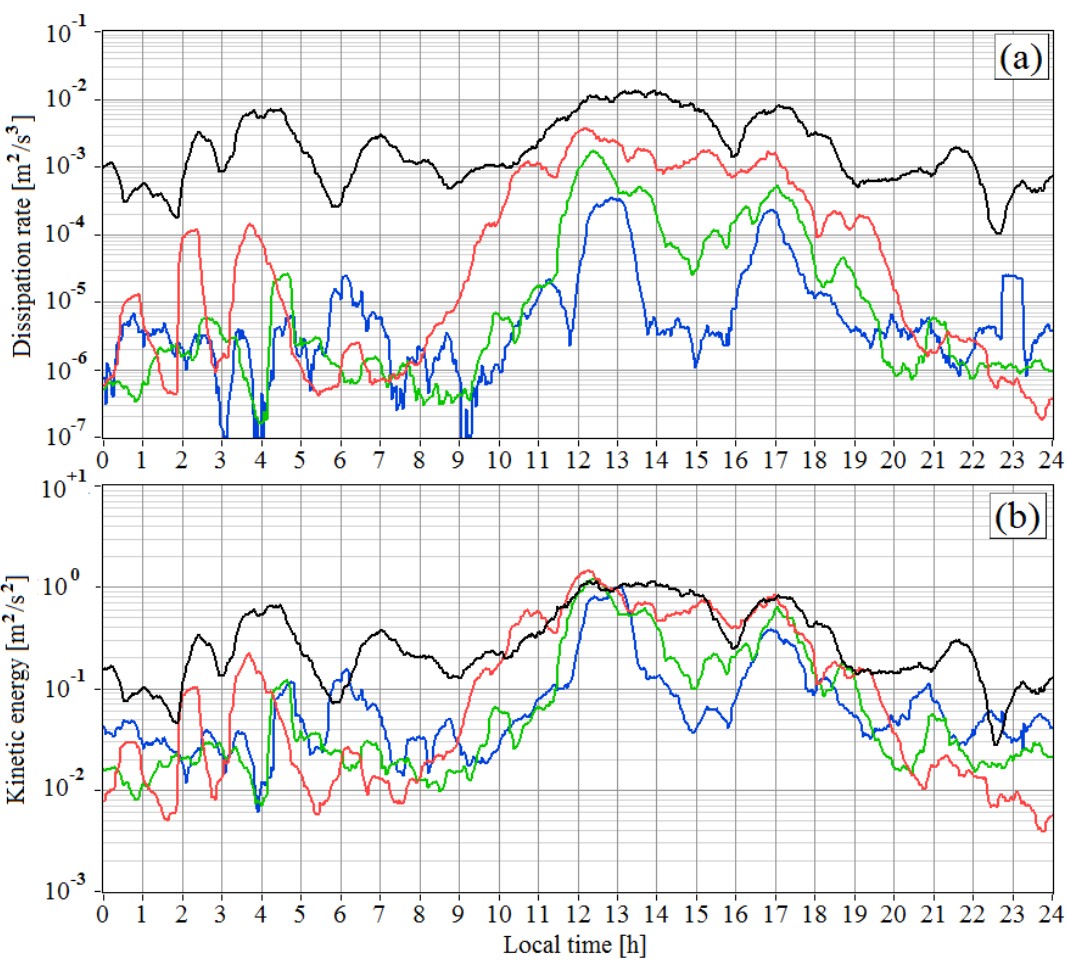

5    **Figure 4**: Time series of the turbulence energy dissipation rate (a) and turbulence kinetic energy (b) at heights of 60 m (black curves), 300 m (red curves), 600 m (green curves), and 900 m (blue curves). Measurements were taken at an elevation angle of 35.3° on July 21 of 2019.



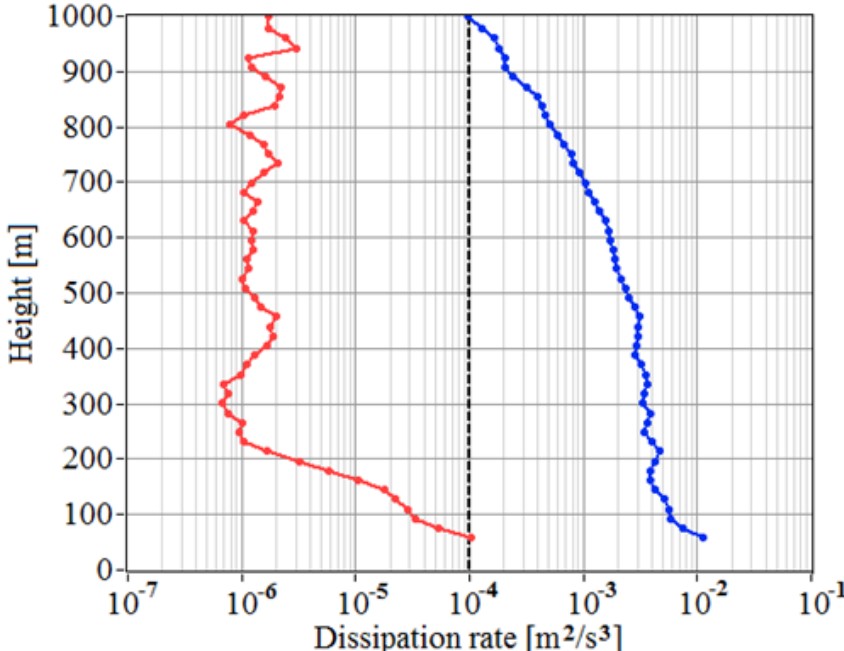

**Figure 5**: Vertical profiles of the turbulence energy dissipation rate obtained from conical scanning by a probing beam around the vertical axis at an elevation angle of 35.3° for 30 min starting from 12:00 LT (blue curve) and 22:30 LT (red curve) on July 21 of 2019. The dashed vertical line corresponds to the threshold $Thr_\varepsilon = 10^{-4}$ m²/s³.



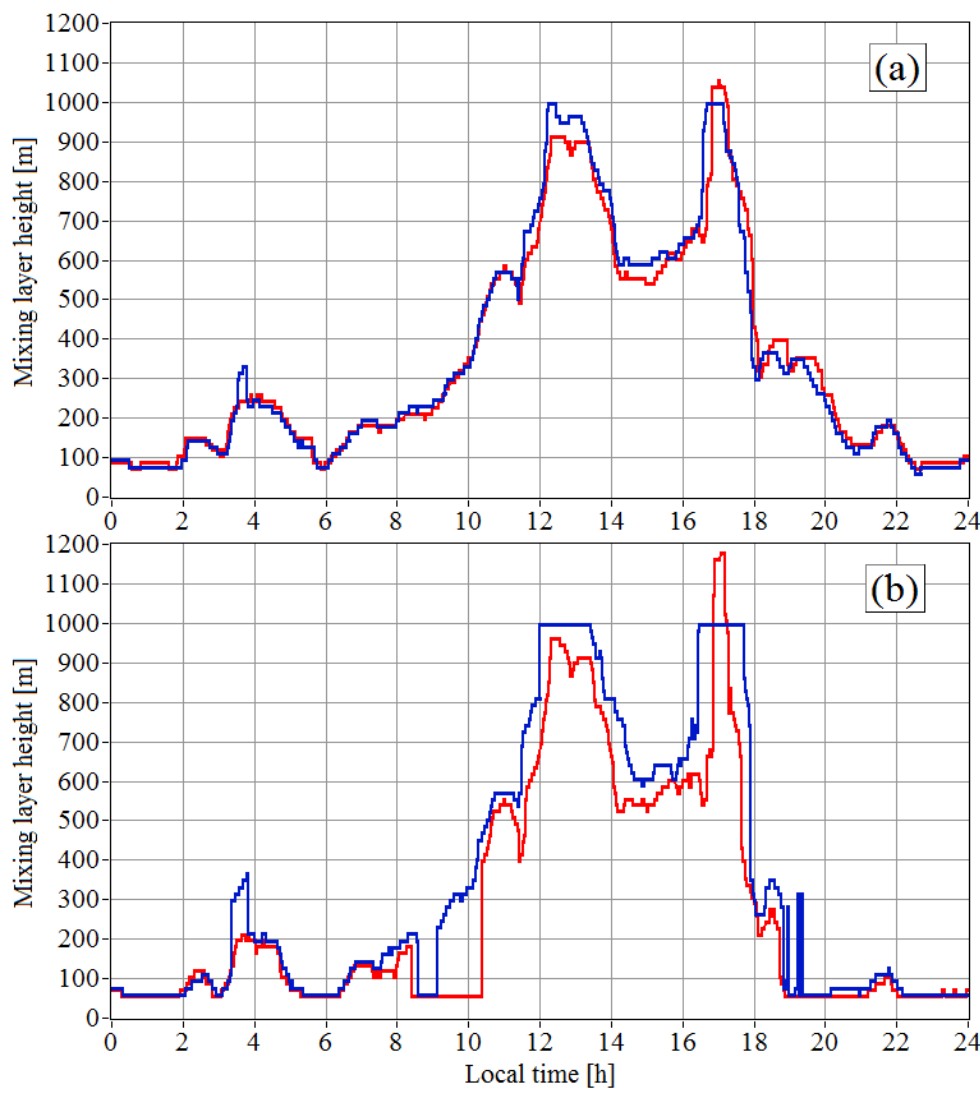

**Figure 6**: Temporal series of the turbulent mixing layer height (thickness) obtained from spatiotemporal distributions of the turbulence
5   energy dissipation rate (a) and the variance of radial velocity (b). Scanning at elevation angles of 60° (red curves) and 35.3° (blue curves).
Measurements were taken on July 21 of 2019. The data of Figs. 2 and 3 are used.

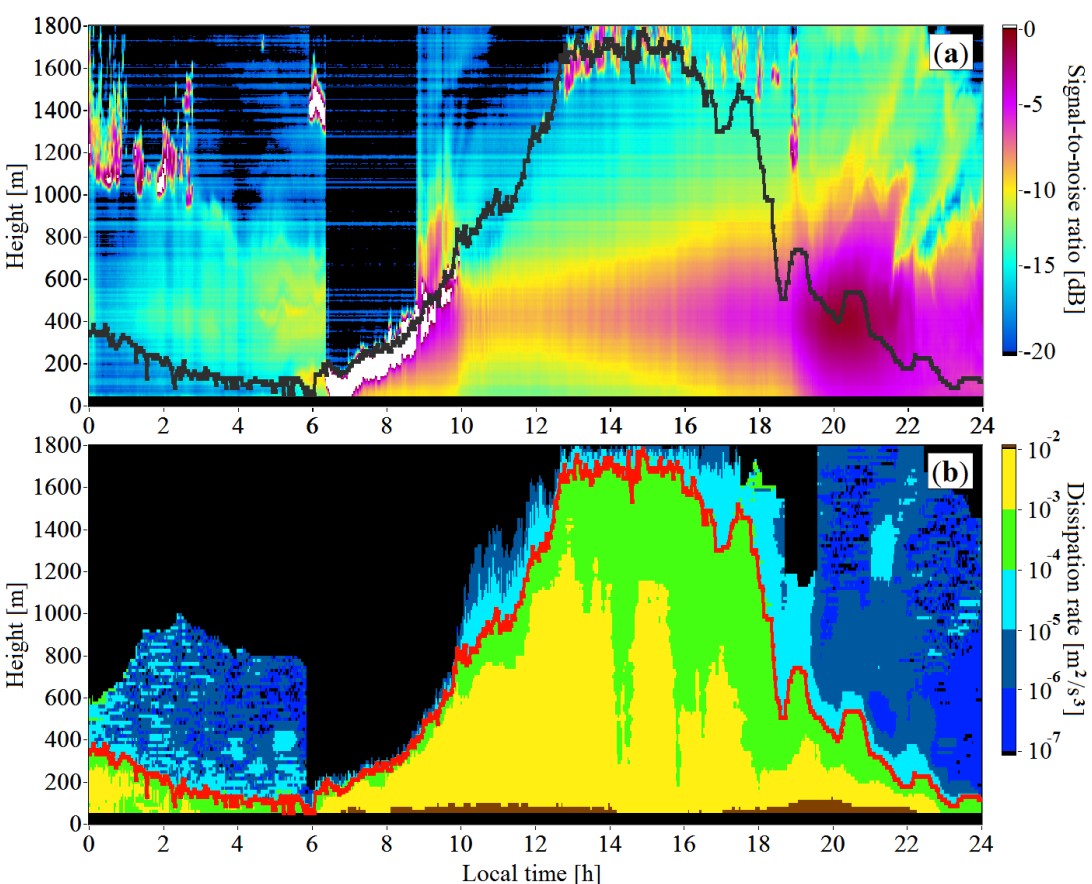

**Figure 7**: Height–temporal distributions of the signal-to-noise ratio (a) and the turbulence energy dissipation rate (b) obtained from
measurements on July 20 of 2019. The gray and red curves show the temporal series of the turbulent mixing layer height assessed from the
equality $\varepsilon(h_{mix}) = 10^{-4}$ m$^2$/s$^3$.


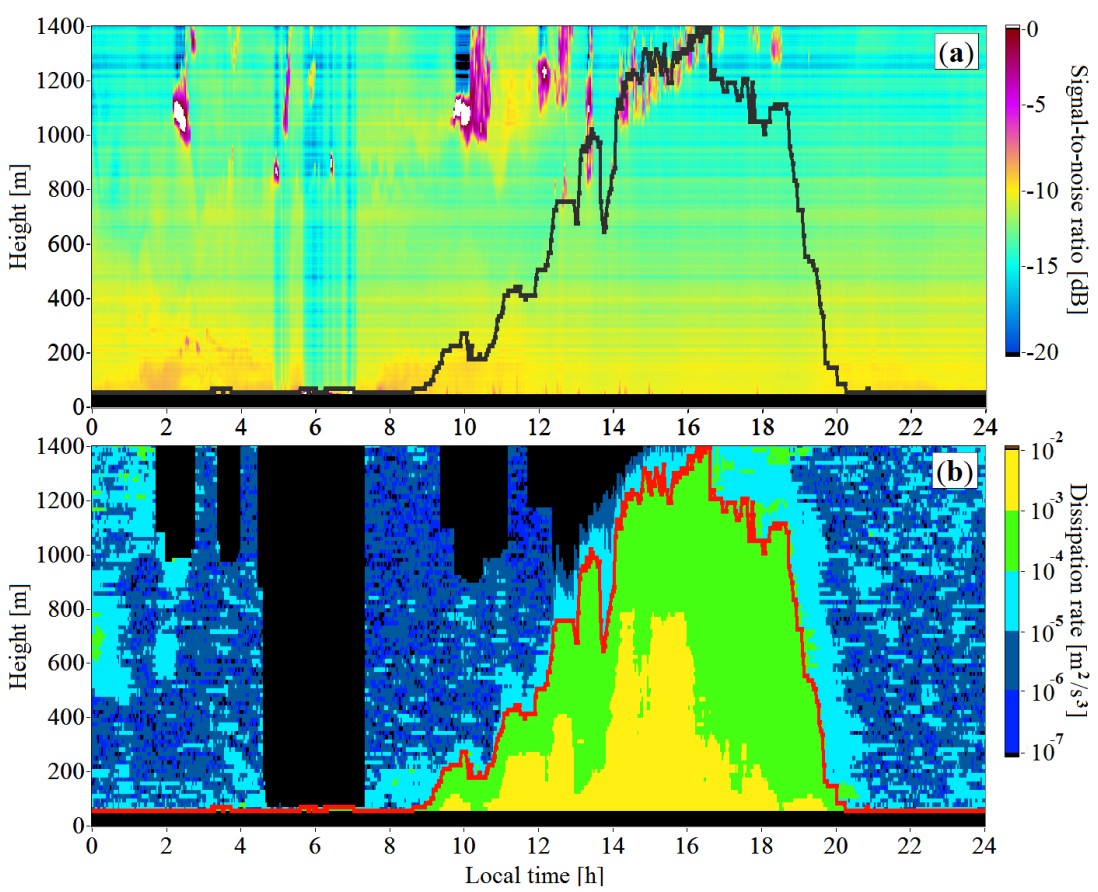

**Figure 8**: Height and time distributions of the signal-to-noise ratio (a) and the turbulence energy dissipation rate (b) obtained from measurements on July 23 of 2019. The turbulent mixing layer height time series is shown by the gray and red curves.





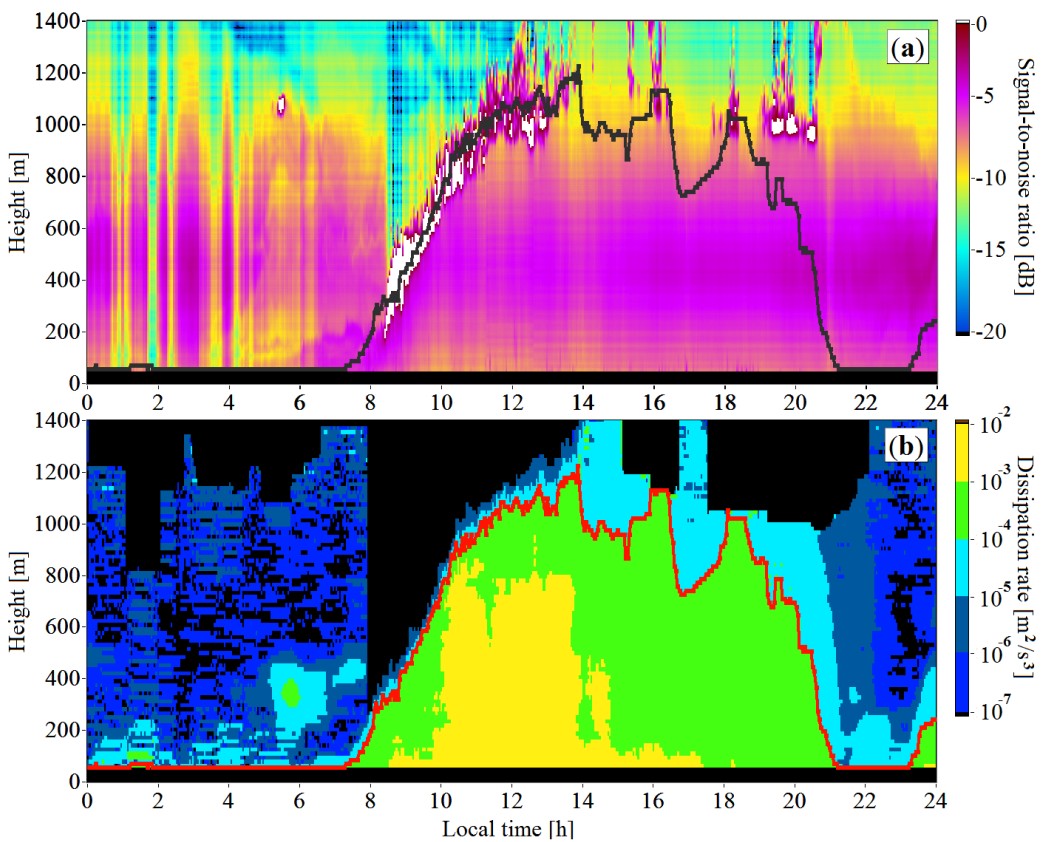

**Figure 9**: Height and time distributions of the signal-to-noise ratio (a) and the turbulence energy dissipation rate (b) obtained on July 25 of 2019. The gray and red curves are for the time series of the turbulent mixing layer height assessed from the data for the dissipation rate.



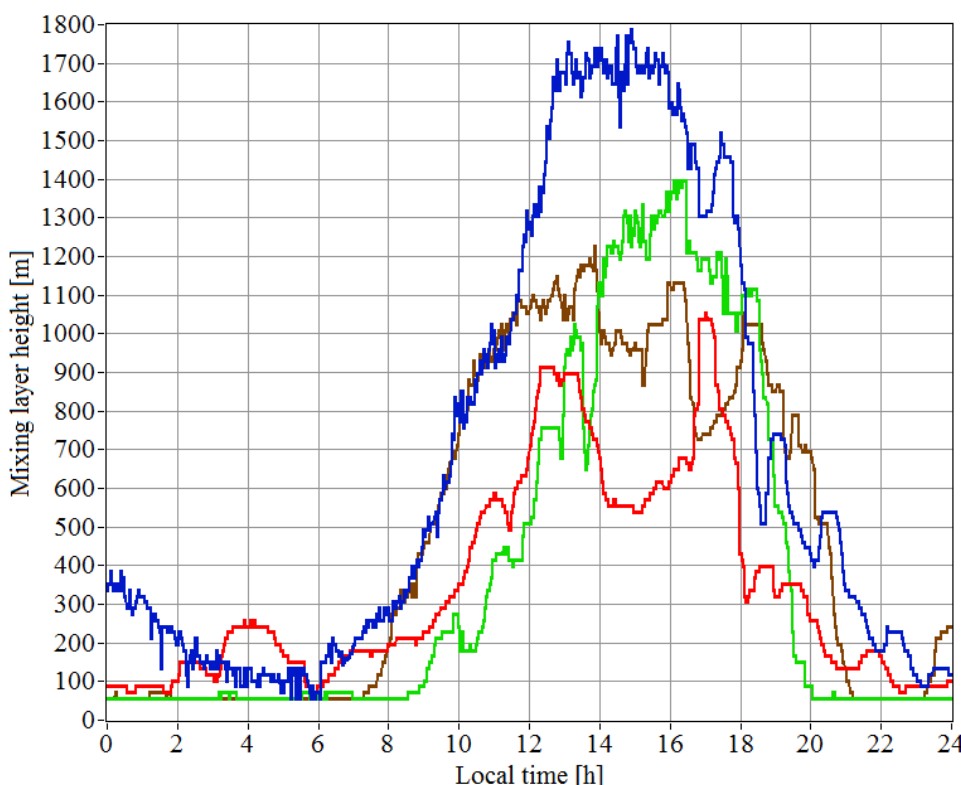

**Figure 10**: Time series of the turbulent mixing layer height (thickness) obtained from lidar data on July 20 (blue curve), 21 (red curve), 23
(green curve), and 25 (brown curve) of 2019.





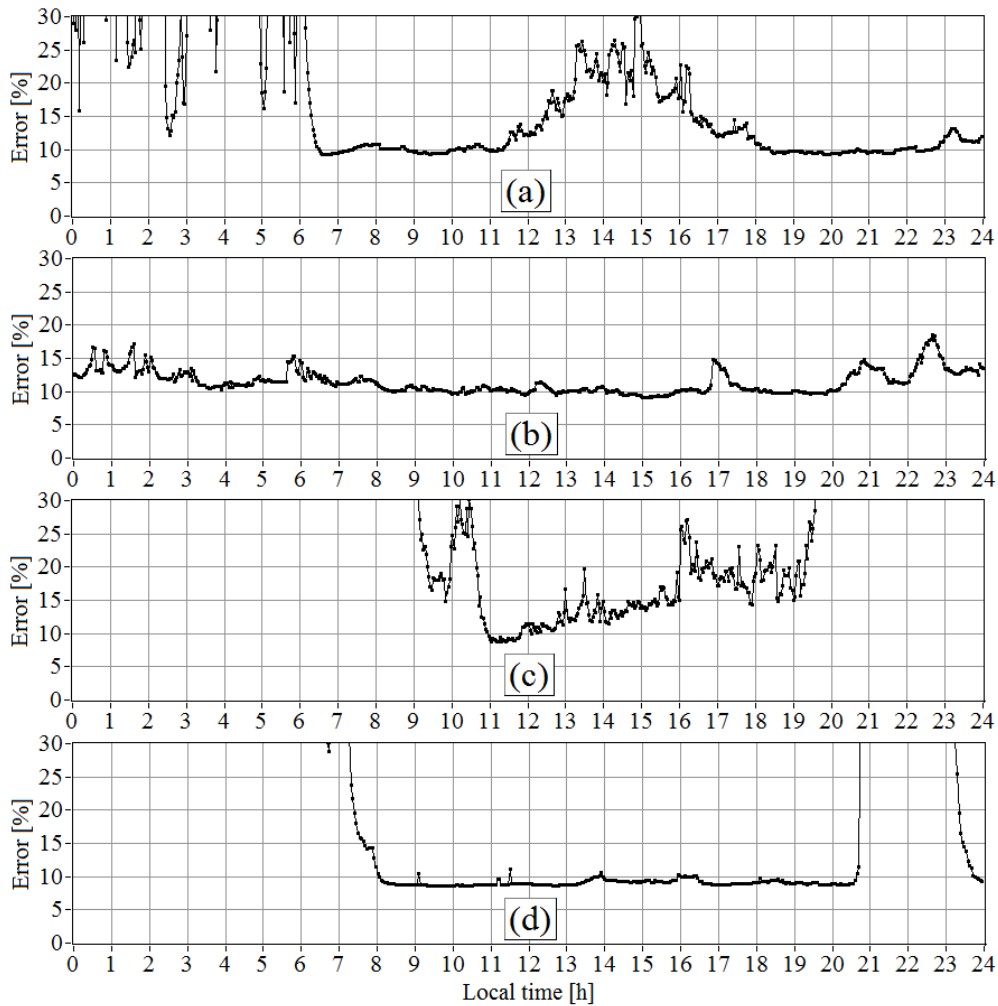

**Figure 11**: Relative error of estimation of the turbulence energy dissipation rate at the mixing layer heights determined from measurements on July 20 (a), 21 (b), 23 (c), and 25 (d) of 2019.





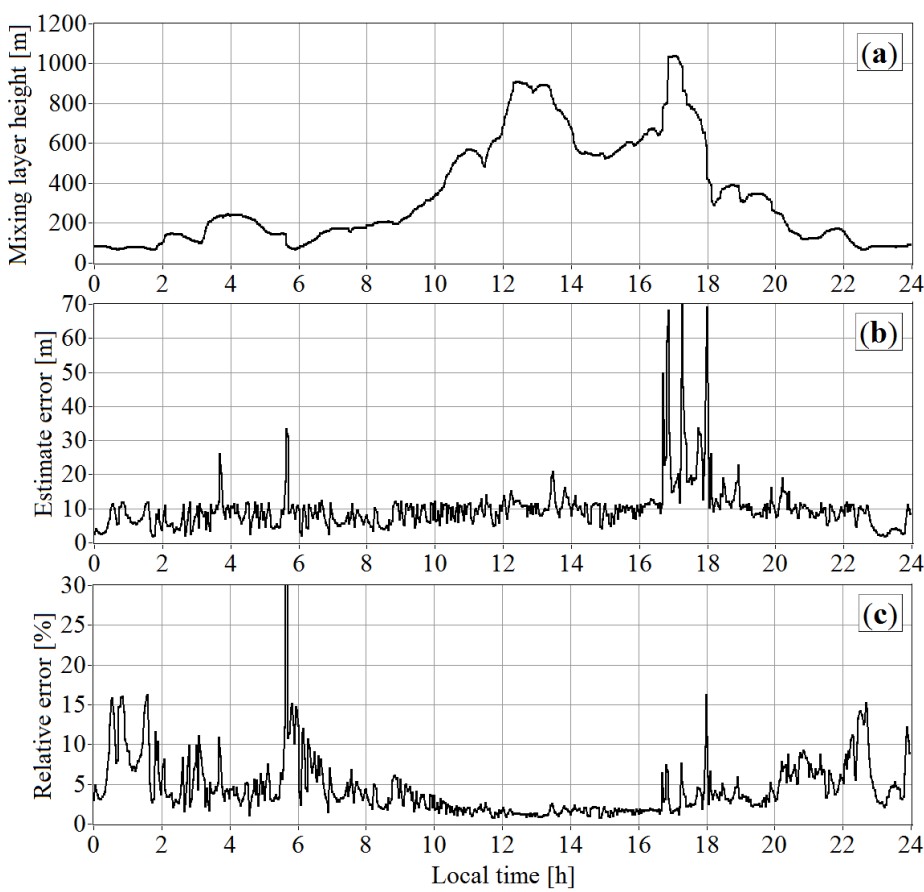

**Figure 12**: The time series of the height of the turbulent mixing layer (a) and the standard (b) and relative (c) errors of its estimation for measurements on July 21 of 2019.



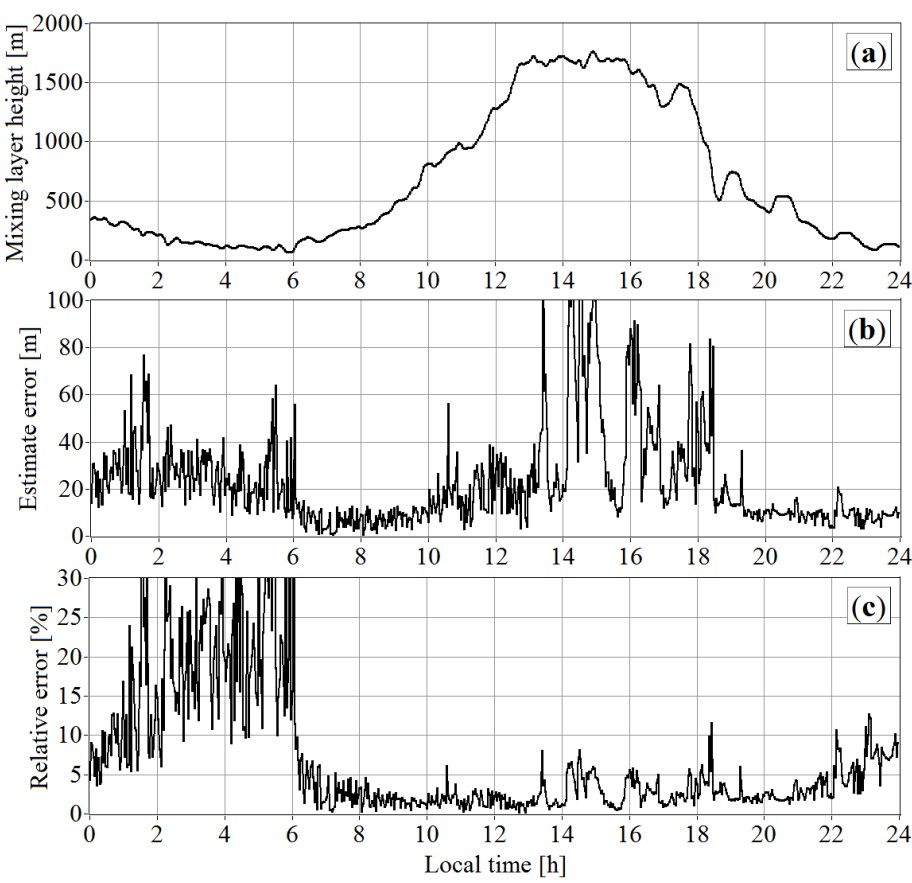

**Figure 13**: The time series of the height of the turbulent mixing layer height (a) and the standard (b) and relative (c) errors of its estimation for measurements on July 20 of 2019.

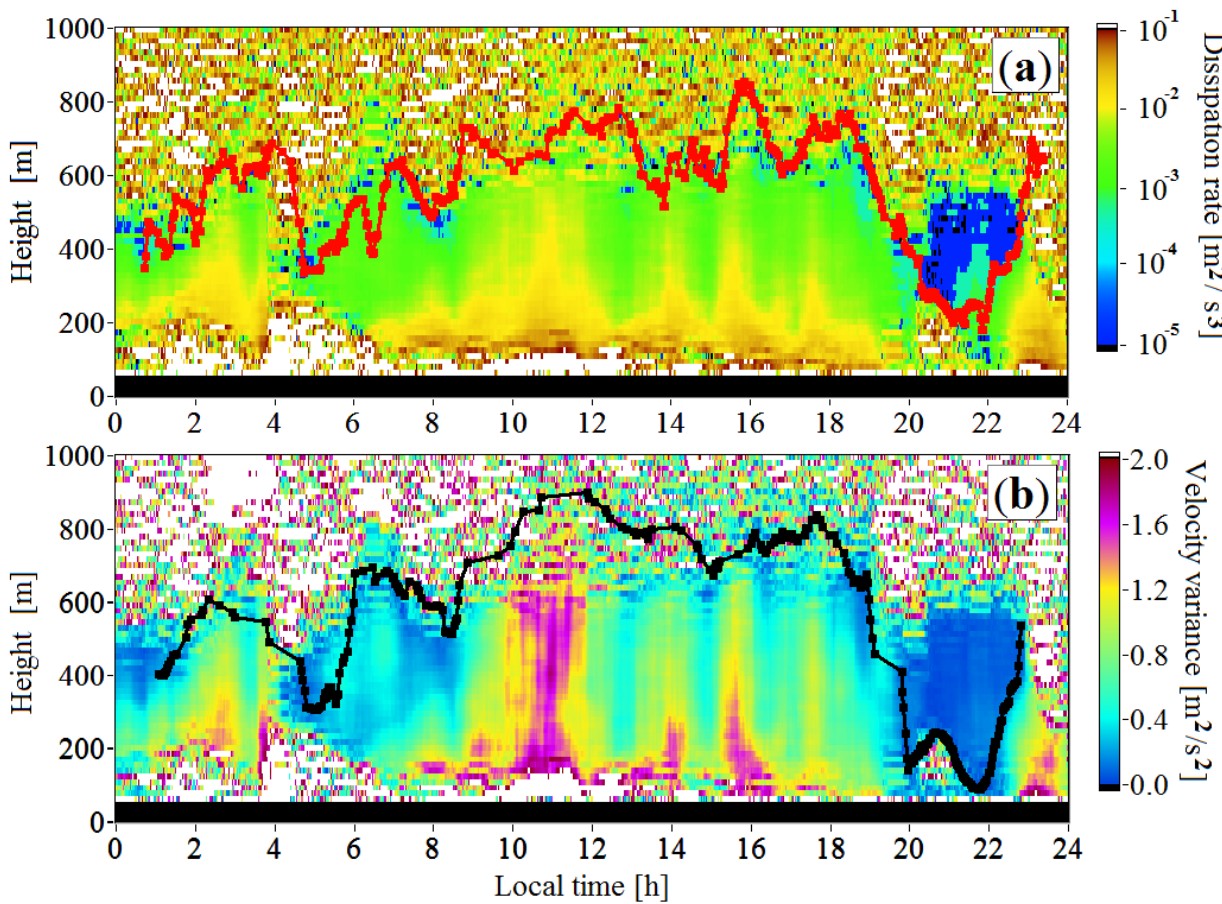

5    **Figure 14**: Height–temporal distributions of the turbulence energy dissipation rate (a) and the radial velocity variance (b) as obtained from lidar measurements on May 1 of 2020. Orange (a) and black (b) curves show the diurnal time series of the turbulent mixing layer height, as estimated from the turbulence energy dissipation rate (orange) and from the variance of radial velocity (black).

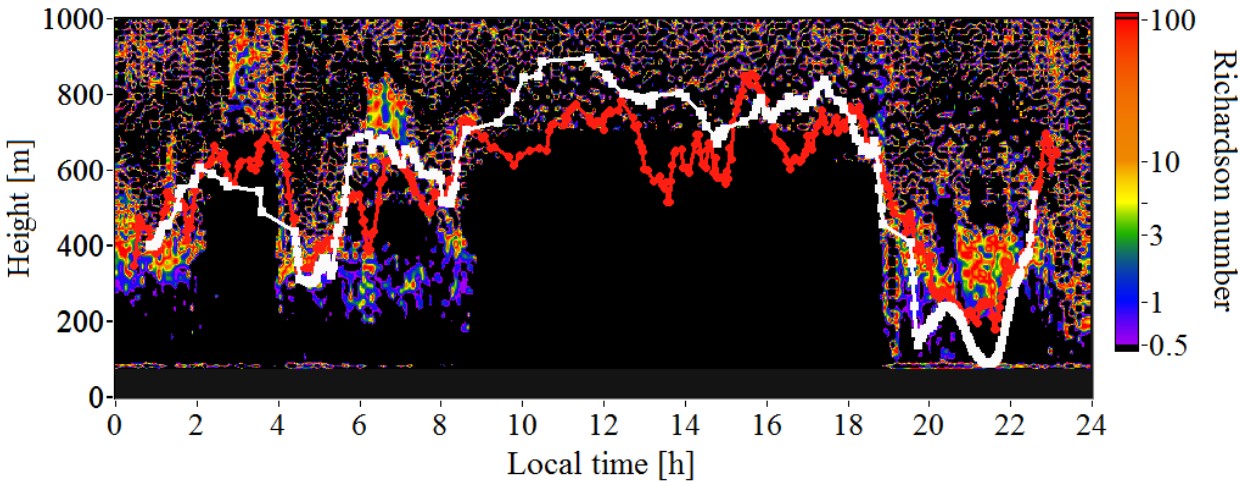

5   **Figure 15**: Height–temporal distribution of the Richardson number as obtained from measurements on May 1 of 2020. Red and white curves  reproduce the diurnal time series of the turbulent mixing layer height, as estimated from the turbulence energy dissipation rate (red) and from the variance of radial velocity (white) which are depicted in Fig.14.