# Peer review of "Estimation of the height of turbulent mixing layer from data of Doppler lidar measurements using conical scanning by a probe beam"

_Atmospheric Measurement Techniques, 2020_

## Referee Comment (RC1) · Anonymous Referee #2 · 28 Sep 2020

This manuscript presents the method for determining the MLH using the dissipation rate of turbulent energy estimated from scanning lidar measurements. The application and accuracy of the method are demonstrated in an experiment in which the wind velocity turbulence was estimated in smog conditions due to forest fires in Siberia in 2019. The results are also validated by comparing to that retrieved from the radial velocity variance and the Richardson number.

The method is useful in mixing layer research. The accuracy of the method is discussed and the analysis is careful. The manuscript is recommended to be published on AMT after major revision, as below:

1. In Sec.3, The details of the turbulence energy dissipation rate could be more briefly.

2. In page 6 line19. The turbulence intensity decreases during 14:00-16:00, as show in Fig. 2 and Fig. 3. The corresponding MLH is obvious low during this period, as show Fig. 6. This may be caused by temperature or the cloud out of detection of lidar. I would recommend removing "On this day, there were no clouds" or add a discussion.

3. In page 14, line 26. I would suggest pasting the details of the corresponding cases, such as SNR, vertical profiles and turbulence intensity, in Appendix. The result of  $35.3^{\circ}$  is recommended to extend to 1.8 km with data quality control.

4. In page 7, line 4 and line 11. "Figs. 2(a) and 2(b)" should be "Figs. 2a and 2b", to keep the format consistent.

5. The journal title abbreviation should be checked, such as in page 17 line 18," Opt. Expr.", in page 16 line 25, "Boundary-Layer Meteorol".

6. Page 1, line 23. "radioacoustic" should be "radio acoustic".

7. Page 3, line 28. "depend" should be "depends".

8. Page 14, line 20-22. Page 15, line 4-7. The descriptions are repetitive.

9. Page 18-20, figure 1-3. The results are both on July 21 of 2019. Suggest plotting in one figure for finding some relationships.

10.Page 25-26, figure 8-9. The two cases are partly cloudy as well as figure 7. Some descriptions about the three cases are repetitive. Also, the two cases are not used for analyzing the relative error of the MLH. What role of the two cases in this manuscript? I would suggest moving to the Appendix.

11.Page 31-32, figure 14-15. The results are both on May 1 of 2020. Suggest plotting in one figure for intercomparing. The data between 800-1000m seems noisy. Do you have data quality control of the raw data? Which line represents the MLH retrieved from the Richardson number in figure 15?

---

## Referee Comment (RC2) · Anonymous Referee #1 · 29 Sep 2020

**Review of 'Estimation of the height of turbulent mixing layer from data of Doppler lidar measurements using conical scanning by a probe beam' by Banakh et al.**

September 28, 2020

The study proposes a method for the determination of the mixing layer height based on profiles of dissipation rate estimated from scanning Doppler lidar measurements. The authors analyze several days from a period with smog related to forest fires. They give estimates of the uncertainty of the method and find that the relative error of the mixing layer height is less than 20 % when the signal-to-noise ratio is high enough and turbulence is sufficiently strong. Finally, the authors compare the mixing layer height to the Gradient Richardson number determined from Doppler lidar and microwave radiometer.

The determination of mixing layer height from dissipation rate itself is not new and has been published in other studies (e.g. Vakkari et al., 2015; Manninen et al., 2018), but the method described here is based on an estimation of dissipation rate developed by the same authors which has not been used to estimate mixing layer heights so far. In addition with the error estimate of mixing layer height this makes an interesting study. However, I have a major concern about the representativity and significance of the conclusions the authors draw. The conclusions are based on very few days only (4 days for the mixing layer height estimate, 2 days for the error estimate and 1 day for the comparison with temperature profiles). The authors mention that data are available for 10 days during a period with forest fires in 2019 and for nearly one month in spring 2020 and I wonder why they do not include data from all available days in their analysis. In my opinion, the results would be much more significant and relevant for the community if they were based on longer time periods and obtained in a objective and statistical way. The statement "It is shown that for the estimation of the mixing layer height (MLH) with the acceptable relative error not exceeding 20 %, the 10 signal-to-noise ratio should be no less than -16 dB, when the relative error of lidar estimation of the dissipation rate does not exceed 30%." is based on manual and subjective analysis of the individual days—at least this is the way it is presented in the manuscript. Besides this major concern, I have several other comments given below.

Based on my evaluation, I cannot recommend the manuscript for publication in AMT in its present state. However, I believe that the study could be suitable for publication, if the authors provided a major revision in which they base their conclusions on an objective and statistical analysis including all available days.

**1 General comments**

1. This comment relates to my major concern described above. Instead of basing the conclusions on error estimates and comparison with temperature profiles on manual evaluation of individual days, the authors should perform an objective and statistical analysis of all available days. This would e.g. make

their recommendation about what SNR values should be used to obtain reliable mixing layer heights estimate more convincing. Instead of showing timeseries and time height sections for individual days (which makes the number of figures unnecessarily long in my opinion) they could show scatter plots, e.g. relate SNR to the error of $\epsilon$ and calculate some statistical measures.

2. Sections 4 and 5 are quite long and confusing. It would be good to include some subsections, e.g. to distinguish the description of the method from the results in Sect. 5.

3. I recommend that the manuscript gets checked by a native speaker before publication.

**2  Specific comments**

1. p. 1, l. 18: "moisture, small gas constituents, pollutants, and heat"

2. p. 1 l. 20-21: The definition of the ABL is fundamental knowledge and one of the classical textbooks such as Stull (1988) or Garratt (1994) should be cited for that.

3. p. 1, l.24: Mixing layer height and ABL should be put into context.

4. p. 1, l. 28: and dissipation rate

5. p. 2, l. 6-7: The recent study of Manninen et al. (2018) should be mentioned here as well.

6. p. 2, l. 7-8: What is meant by vertical scanning? What is the difference compared to vertical stare mode and conical scanning?

7. p. 2, l. 17: Here and throughout the manuscript: The authors define abbreviatons e.g. $h_{mix}$ or $\epsilon$, but do not use them consistently throughout the manuscript. Instead the sometimes use the long name or both. Once an abbreviation is introduced it should be used consistently.

8. p. 3, l.3ff: The motivation and objectives of the study should be made clearer. What is new compared to previous work? How are the objectives addressed? What data are used?

9. p. 3, l. 23: Not clear what the scan number is. What is a scan? Full azimuth scan of 360 degree at a certain elevation angle?

10. p. 4, l. 12: Isn't $D_L$ still a function of $R_k$?

11. p. 4, l. 21: What is $y$ in $A(y)$?

12. p. 4, l. 22ff: Thus, $\overline{\sigma}_r^2$ depends on $\epsilon$. How does that effect the comparison of mixing layer height determined from both quantities?

13. p. 5, l. 12-13: How strongly does this threshold vary in literature? Did the authors investigate how sensitive their results to the chosen threshold are?

14. p. 5, l. 16ff: Why did the authors choose the period during wild fires for the analysis? What impact on the ABL conditions and their method do they expect? What are the site characteristics, i.e. terrain, surface conditions, ...? Are there other instruments deployed simultaneously? Later they mention surface flux measurements.

15. p. 5, l. 22: Why are the accumulation numbers used for the different elevation angles different?

16. p. 6, l. 5ff: How is the threshold of -15 dB for SNR determined? How is the relative error of 30 % for turbulence parameters determined? How do clouds and fog affect the measurements in clear parts of the atmosphere? Please explain. Given that the whole time period encompasses 10 days only, it could be interesting to show 10-day time height sections of SNR and wind for the whole period to get an overview and information on the variety of atmospheric conditions.

17. p. 6, l. 12: What is meant by 'the elevation angle was alternated for $\Delta\tau \approx 1.5s$? Until here, I was assuming that consecutive full azimuth scans were done one after the other at each elevation angle.

18. p. 6, l. 23: An objective comparison e.g. by calculating RMSE and correlation coefficient, would be more meaningful than stating that there are practically no differences.

19. p. 7, l. 2-3: 'sufficiently high signal-to-noise ratio': how is that determined? What is the criterion to distinguish between sufficiently high and low SNR?

20. p. 7, l. 4ff: Like above an objective comparison by calculating RMSE and correlation coefficient would be more meaningful. Also, the comparison should be done for the whole period to make the result more meaningful.

21. p. 7, l. 14-15: This is interesting. Do the authors have any hypothesis why TKE was similar while dissipation rate decreased with height?

22. p. 7, l. 17-18, Fig. 5: The profiles plot do not really show any new information compared to the time height sections. What is the purpose of including them?

23. p. 7, l. 22: Where does the threshold used for the radial velocity variance profiles come from? Is that based on other literature? Is it empirical?

24. p. 7, l. 25: In case the values of $\sigma_r$ or $\epsilon$ are smaller than their corresponding threshold, the mixing layer height is set to 60 m. Why not set it too missing? It is well possible that no mixing layer exists at all.

25. p. 7, l. 28: similar like to the comment about the sufficiently high SNR above. Which objective criteria is used to determine low quality data?

26. p. 8, l. 3-4: An objective comparison between mixing layer height estimates for the different elevation angles could easily be done for all available days.

27. p. 8, l. 8ff: What is the purpose of showing all these examples with plots of dissipation rate and SNR profiles? Like outlined in my major comment, an objective and statistical comparison would make the results more meaningful. The example plots for the individual days could e.g. be put in a supplemental or appendix.

28. p. 8, l. 17-18: Can this change in SNR be related to a change in wind direction which could explain the enhanced advection of smog?

29. p. 8, l. 19-21: How reliable is $\epsilon$ calculated in the cloud layer? The mixing layer height is determined at the top of the layer with high SNR, i.e. somewhere in the lower part of the cloud. Depending on the cloud characteristics mixing may reach up much further. Thus, the estimated height does not necessarily agree with the true mixing layer height but is simply an affect of how deep the lidar beam penetrates into the cloud.

30. p. 9, l. 14: Please explain the difference between probing volume (30 m) and range gate length (18 m).

31. p. 9, l. 17-18: The fact that cloud base and the detected mixing layer coincide does not confirm the correctness of the method. As seen in the examples, mixing layer heights are detected at the top of the layer with maximum SNR, i.e. in the lower part of the cloud. Mixing layer heights may be deeper. The correctness of the method can only be confirmed by comparing it to independent reliable measurements, such as radiosonde profiles.

32. p. 9, l. 24: It would make it easier to understand how the relative error of $\epsilon$ is calculated, if the equation was given.

33. p. 9, l. 26: Why only look at the relative error at mixing layer height? What is the time height section of this error?

34. p. 9, l. 27ff, Fig. 11: In my opinion, Fig. 11 is not ideal and the explanation why the error is high or low could be much easier to follow if a scatter plot between the error and SNR was shown.

35. p. 10, l. 21ff: What is the purpose of the series of closed numerical experiments? It is not clear how they are done. What are the preset values of the mixing layer height?

36. p. 11, l. 1ff: What is the experimental error? On what assumption is the threshold of 30 % for the relative error based?

37. p. 11, l. 14ff: While reading this paragraph I was wondering how the calculations are done. This information is given in the following paragraph and I suggest changing the order.

38. p. 11, l. 25: What do the random realizations of $\epsilon$ look like? How much do they differ from the original profiles? It could be interesting to show some profiles.

39. p. 12, l. 4ff: the error for mixing layer height obtained with the described method is very small. A discussion of other uncertainties related to the mixing layer height, e.g. the sensitivity to the used threshold, should be included. To really assess how correct the determined mixing layer heights are, comparison with independent measurements such as radiosoundings would be necessary. It would be interesting to see if the mixing layer height determined from $\epsilon$ would agree with mixing layer heights determined from other instruments (Emeis et al., 2008) within the uncertainty range.

40. p. 12, l. 22: The Richardson number describes if turbulence can develop in a stably stratified atmosphere (e.g. Stull, 1988). The static stability is described by the temperature gradient.

41. p. 13, l. 4ff: It should be mentioned in the beginning that an additional data set from a period in 2020 is used. Also, if data from a microwave radiometer are used much more information on this instrument needs to be given. The temperature profile retrieved from the passive instrument are prone to uncertainties and errors and information on its accuracy and the used retrievals should be given. Microwave radiometers often struggle to resolve elevated inversion at the top of the ABL and thus the gradient Richardson number obtained from these instruments have to be used with care and it should not be taken as granted that the correctly detect the inversion at the top of the ABL and that they can be used to validate the mixing layer height detected from $\epsilon$.

42. p. 13, l. 22ff: Like above, a statistical objective analysis of the whole period should be conducted and the conclusion that the mixing layer height derived from $\epsilon$ agrees well with the gradient Richardson number using a threshold of 0.5 should be based on the whole data set and not just on a single example day.

43. p. 14, l. 5: A mixing layer height of several hundred meters during the night must be shear driven. A discussion of the physical processes causing the mixing layer is missing and should be added to the pure description of the profiles.

44. p. 14, l. 20: Second period in 2020 should be mentioned.

45. p. 15, l. 13-14: It is not clear to me where the vertical gradient of $\epsilon$ is considered in the error estimate.

46. p. 15, l. 14: The result 'SNR should be no less than -16 dB' is not clear to me. On what analysis is that based?

47. p. 15, l. 16ff: A good way of showing this could be by plotting the error of the mixing layer height over $\epsilon$.

**References**

Emeis, S., Schafer, K., and Munkel, C.: Surface-based remote sensing of the mixing-layer height - a review, 17, 621–630, doi:doi:10.1127/0941-2948/2008/0312, 2008.

Garratt, J. R.: The atmospheric boundary layer, Cambridge atmospheric and space science series, Cambridge University Press, 316 pp, Cambridge, 1994.

Manninen, A., Marke, T., Tuononen, M., and O'Connor, E.: Atmospheric boundary layer classification with Doppler lidar, J. Geophys. Res., 123, 8172–8189, doi:10.1029/2017JD028169, 2018.

Stull, R. B.: An introduction to boundary layer meteorology, Kluwer Academic Publishers, 666 pp, Dordrecht, The Netherlands, 1988.

Vakkari, V., O'Connor, E., Nisantzi, A., Mamouri, R., and Hadjimitsis, D.: Low-level mixing height detection in coastal locations with a scanning Doppler lidar, Atmos. Meas. Tech., 8, 1875–1885, doi:10.5194/amt-8-1875-2015, 2015.

---

## Author Comment (AC1) · 28 Oct 2020

Reply to the Reviewer We thank very much the Reviewer for his time and efforts, thoughtful and very useful comments. We will re-arrange material and incorporate the most part of suggested revisions to the manuscript, if the Editor give us a chance to revise it. Authors

---

## Author Comment (AC2) · 28 Oct 2020

Reply to the Reviewer We thank very much the Reviewer for his time and efforts, thoughtful and very useful comments. We will re-arrange material and incorporate the most part of suggested revisions to the manuscript, if the Editor give us a chance to revise it. In addition. Explanation, why we restricted the number of considered examples of the experiment in July 2019 is listed on page 6, lines 4-5 of the manuscript. As to the experiment in spring 2020, the weather was stormy: snowy, fog, rainy. Nevertheless, we will add the examples of comparison with the Richardson number in the periods when impact of these factors was minimal. Authors

---

## Author Comment (AC4) · 9 Nov 2020

[revised manuscript text omitted]

**Reply to the Reviewers of the manuscript**

We thank very much the Reviewers for their time and efforts, thoughtful and very useful comments. We have incorporated the most of their suggested revisions as indicated below. Changes and additions in the revised manuscript are marked by yellow.

**Reviewer 1**

This manuscript presents the method for determining the MLH using the dissipation rate of turbulent energy estimated from scanning lidar measurements. The application and accuracy of the method are demonstrated in an experiment in which the wind velocity turbulence was estimated in smog conditions due to forest fires in Siberia in 2019. The results are also validated by comparing to that retrieved from the radial velocity variance and the Richardson number.

The method is useful in mixing layer research. The accuracy of the method is discussed and the analysis is careful. The manuscript is recommended to be published on AMT after major revision, as below:
1. In Sec.3, The details of the turbulence energy dissipation rate could be more briefly.

Fixed. Figs. 4, 5 of the initial version of the manuscript are removed as well as comments to these Figures. Since Section 3 describes the experiment, we have changed the title of this section "Evaluation of the turbulent mixing layer height during forest fires in Siberia in 2019" to "Experiment during forest fires in Siberia in 2019".

2. In page 6 line19.The turbulence intensity decreases during 14:00-16:00, as show in Fig. 2 and Fig. 3. The corresponding MLH is obvious low during this period, as show Fig. 6. This may be caused by temperature or the cloud out of detection of lidar. I would recommend removing "On this day, there were no clouds" or add a discussion.

Page 7, lines 1-2: The sentence "First, we will consider the results of lidar measurements in a cloudless atmosphere (at least up to height of 1800 m) and at the highest signal-to-noise ratio" has been added. In the same paragraph, the sentence " On this day, there were no clouds, and the smog was observed from 00:00 to 24:00 Local Time up to a height of at least 2 km." is deleted.
We assume that the decrease in the turbulence intensity during 14:00-16:00 is caused by a rapid change in wind direction (see Figure 1c) during that period. At least the dependence of the height of the turbulent mixing layer, shown in Figure 4a, is in agreement with the changes in the height profiles of the wind direction angle in the time period from 10:00 to 19:00.

3. In page 14, line 26. I would suggest pasting the details of the corresponding cases, such as SNR, vertical profiles and turbulence intensity, in Appendix. The result of 35.3° is recommended to extend to 1.8 km with data quality control.
We cannot increase the maximum height to 1800 m when measuring at an elevation angle of 35.3°, since during the experiment we set the maximum range of 2100 m (3000 m).
Page 6, lines 5-8: The sentence "At the beginning of the experiment, we set the maximum ... at elevation angles of 35.3° and 60°, respectively)." has been added.

4. In page 7, line 4 and line11. "Figs. 2(a) and 2(b)" should be "Figs. 2a and 2b", to keep the format consistent.

Fixed.

5. The journal title abbreviation should be checked, such as in page 17 line 18,” Opt. Expr.”, in page 16 line 25, ”Boundary-Layer Meteorol”.

Fixed.

6. Page 1, line 23. "radioacoustic" should be "radio acoustic".

Fixed.

7. Page 3, line 28. "depend" should be "depends".

The instrumental error and the probability depend on SNR ...

8. Page 14, line 20-22. Page 15, line 4-7. The descriptions are repetitive.

The sentence "The experiments were carried out in the territory of the Basic Experimental Observatory of the Institute of Atmospheric Optics in Tomsk with the use of the Stream Line lidar (Halo Phtonics, Brockamin, Worcester, United Kingdom)." and the text "On July 20 and 25, the time series of MLH are practically identical during the period from 8:00 to almost 12:00. In both cases, MLH was increasing in this period, and its increase was accompanied by the rise of the cloud base due to convection. This confirms the correctness of the MLH time series assessment based on the height–temporal distributions of the turbulence energy dissipation rate." have been removed.

9. Page 18-20, figure 1-3. The results are both on July 21 of 2019. Suggest plotting in one figure for finding some relationships.

We think that combining Figures 1-3 into one figure will be inconvenient for the reader of this paper. Too much information for one figure.

10. Page 25-26, figure 8-9. The two cases are partly cloudy as well as figure 7. Some descriptions about the three cases are repetitive. Also, the two cases are not used for analyzing the relative error of the MLH. What role of the two cases in this manuscript? I would suggest moving to the Appendix.

We removed Figures 8 and 9 (figure numbering in the original version of the manuscript), but added Figure 5 (figure numbering in the revised manuscript), where there is information about the signal-to-noise ratio, wind and dissipation rate for 6 days of the experiment.

11. Page 31-32, figure 14-15. The results are both on May 1 of 2020. Suggest plotting in one figure for intercomparing. The data between 800-1000m seems noisy. Do you have data quality control of the raw data? Which line represents the MLH retrieved from the Richardson number in figure 15?

Text on 13 and 14 pages of the initial version of the manuscript is changed (13, 14 pages of the revised manuscript). We added new data and replaced Figs. 14, 15 (initial version) by Fig.11 (revised version) where the dissipation rate and the Richardson number are plotting together. In the height-temporal distributions of the Richardson number as a height of the turbulent mixing layer we took a minimal height above which the Richardson number exceeds 0.5. (Lines 23, 24 on 13 page of the revised manuscript). Yes, we did quality control of the raw data (Lines 3-8 on 13 page of the revised manuscript).

**Reviewer 2**

Review of 'Estimation of the height of turbulent mixing layer from

September 28, 2020

The study proposes a method for the determination of the mixing layer height based on profiles of dissipation rate estimated from scanning Doppler lidar measurements. The authors analyze several days from a period with smog related to forest fires. They give estimates of the uncertainty of the method and find that the relative error of the mixing layer height is less than 20 % when the signal-to-noise ratio is high enough and turbulence is sufficiently strong. Finally, the authors compare the mixing layer height to the Gradient Richardson number determined from Doppler lidar and microwave radiometer.

The determination of mixing layer height from dissipation rate itself is not new and has been published in other studies (e.g. Vakkari et al., 2015; Manninen et al., 2018), but the method described here is based on an estimation of dissipation rate developed by the same authors which has not been used to estimate mixing layer heights so far. In addition with the error estimate of mixing layer height this makes an interesting study. However, I have a major concern about the representativity and significance of the conclusions the authors draw. The conclusions are based on very few days only (4 days for the mixing layer height estimate, 2 days for the error estimate and 1 day for the comparison with temperature profiles). The authors mention that data are available for 10 days during a period with forest fires in 2019 and for nearly one month in spring 2020 and I wonder why they do not include data from all available days in their analysis. In my opinion, the results would be much more significant and relevant for the community if they were based on longer time periods and obtained in a objective and statistical way. The statement "It is shown that for the estimation of the mixing layer height (MLH) with the acceptable relative error not exceeding 20 %, the 10 signal-to-noise ratio should be no less than -16 dB, when the relative error of lidar estimation of the dissipation rate does not exceed 30%." is based on manual and subjective analysis of the individual days at least this is the way it is presented in the manuscript. Besides this major concern, I have several other comments given below.

Based on my evaluation, I cannot recommend the manuscript for publication in AMT in its present state. However, I believe that the study could be suitable for publication, if the authors provided a major revision in which they base their conclusions on an objective and statistical analysis including all available days.

From lidar measurements during the 10-day experiment, data obtained only within 6 days proved to be suitable. Perhaps this amount of data is not enough for full-fledged statistics, but our main goal was to test the proposed method for determining the height of the mixing layer from the vertical profiles of the turbulent energy dissipation rate, retrieved from measurements by the Stream Line lidar with conical scanning, for which the pulse energy is rather low. Due to the low pulse energy under normal conditions, it is possible to retrieve the vertical profiles of wind turbulence parameters (including the dissipation rate) from measurements with this lidar to a maximum height of 500 m, which is not enough to obtain the dependence of the mixing layer height for a full day. From 20 to 27 July 2019, there was smog at the site of the experiment due to forest fires in Siberia and this gave us the opportunity to obtain vertical profiles of turbulence in the entire mixing layer. The method described here can be applied to data measured by a high-power pulsed coherent Doppler lidar under normal conditions (with a background aerosol), which will enable a full-fledged statistical analysis.

We do not fully agree with the Reviewer's criticism of the main conclusions of this work, since they are based on reliable experimental results, despite the relatively small amount of data measured by the lidar. Nevertheless, under revising the manuscript, we extended the experimental data which are considered and analyzed, and tried to take into account comments from the Reviewer maximally.

**1 General comments**

1. This comment relates to my major concern described above. Instead of basing the conclusions on error estimates and comparison with temperature profiles on manual evaluation of individual days, the authors should perform an objective and statistical analysis of all available days. This would e.g. make their recommendation about what SNR values should be used to obtain reliable mixing layer heights estimate more convincing. Instead of showing time series and time height sections for individual days (which makes the number of figures unnecessarily long in my opinion) they could show scatter plots, e.g. relate SNR to the error of the dissipation rate and calculate some statistical measures.

2. Sections 4 and 5 are quite long and confusing. It would be good to include some subsections, e.g. to distinguish the description of the method from the results in Sect. 5.

3. I recommend that the manuscript gets checked by a native speaker before publication.

We see no reason to include in this article the experimental dependences of the error of the dissipation rate estimate on the signal-to-noise ratio, since this issue was investigated earlier in the work by Banakh et al. (2017).

We have significantly revised sections 4 and 5. In particular, the text in these sections was shortened, Figures 4, 5, 7-13 were removed (figures were numbered in the original version of the manuscript), Figures 5-10 were added (figures are numbered in the revised manuscript).

The initial version of the manuscript was proofed by the MDPI English Editing Service and we have Certificate confirming that "the text has been checked for correct use of grammar and common technical terms, and edited to a level suitable for reporting research in a scholarly journal".

**2 Specific comments**

1. p. 1, l. 18: "moisture, small gas constituents, pollutants, and heat"

Page 1, line 18: The text "and pollutants" has been replaced by "pollutants, and heat".

2. p. 1 l. 20-21: The definition of the ABL is fundamental knowledge and one of the classical text books such as Stull (1988) or Garratt (1994) should be cited for that.

The references "Stull (1988)" and "Garratt (1994)" have been added.

3. p. 1, l.24: Mixing layer height and ABL should be put into context.

We are not sure if we understand this reviewer comment correctly.

4. p. 1, l. 28: and dissipation rate

Page 2, line 1-2: The text "and turbulent energy dissipation rate $\varepsilon(h)$" has been added.

5. p. 2, l. 6-7: The recent study of Manninen et al. (2018) should be mentioned here as well.

The references "Manninen et al. (2018)" has been added.

6. p. 2, l. 7-8: What is meant by vertical scanning? What is the difference compared to vertical stare mode and conical scanning?

Page 2, line 10: The text "vertical scanning" has been replaced by "scanning in vertical plane".

7. p. 2, l. 17: Here and throughout the manuscript: The authors define abbreviatons e.g. $h_{mix}$ or $\varepsilon$, but do not use them consistently throughout the manuscript. Instead the sometimes use the long name or both. Once an abbreviation is introduced it should be used consistently.

We try to accommodate this remark.

8. p. 3, l.3ff: The motivation and objectives of the study should be made clearer. What is new compared to previous work? How are the objectives addressed? What data are used?

We try to correct that, lines 6-12, page 3 of the revised manuscript.

9. p. 3, l. 23: Not clear what the scan number is. What is a scan? Full azimuth scan of 360 degree at a certain elevation angle?

Page 3, 4, lines 30, 1: The text "(full azimuth scan of 360 degree at a certain elevation angle)" has been added.

10. p. 4, l. 12: Isn't $D_L$ still a function of $R_k$?

Yes.

11. p. 4, l. 21: What is $y$ in $A(y)$?

Page 5, line 3: "$\Delta y_k = \Delta\theta R_k \cos\varphi$ ($\Delta\theta$ is in radians)" has been added.

12. p. 4, l. 22ff: Thus, $\bar{\sigma}_r^2$ depends on $\varepsilon$. How does that effect the comparison of mixing layer height determined from both quantities?

In equation (5), in addition to the instrumental error of radial velocity estimate, we take into account the averaging of the radial velocity over the probed volume. If the size (longitudinal or transverse) of this volume is less than the integral scale of turbulence, then the last term in Eq. (5) depends only on the dissipation rate. Without taking into account the averaging of the radial velocity over the probed volume, the variance estimate $\bar{\sigma}_r^2$ is underestimated, and therefore, the estimate of the height of the mixing layer from $\bar{\sigma}_r^2(h)$ will also be underestimated.

13. p. 5, l. 12-13: How strongly does this threshold vary in literature? Did the authors investigate how sensitive their results to the chosen threshold are?

We use the thresholds corresponding to the lower limit of moderate turbulence.

14. p. 5, l. 16ff: Why did the authors choose the period during wild fires for the analysis? What impact on the ABL conditions and their method do they expect? What are the site characteristics, i.e. terrain, surface conditions, ...? Are there other instruments deployed simultaneously? Later they mention surface flux measurements.

We use a Stream Line lidar with a rather low pulse energy and therefore, from measurements under normal conditions (at the place of the experiment, as a rule, a low aerosol concentration takes place), we can determine the parameters of wind turbulence up to an height of no more than 500 m. Since this is not enough to study the mixing layer height, we have chosen the period when the aerosol concentration (and therefore SNR) is high enough due to forest fires.

15. p. 5, l. 22: Why are the accumulation numbers used for the different elevation angles different?
We used the same accumulation number for the different elevation angles. But the accumulation numbers were different on different days.
Page 5, 6, lines 26, 1: "(until 12:30 22 July 2019)" and "(after 12:30 22 July 2019)" have been added.
Page 6, lines 5-8: The sentence "At the beginning of the experiment, we set the maximum range $R_{K-1}$ equal to 2100 m (maximum measurement heights $h_{K-1}$ of 1213 m and 1818 m at elevation angles of 35.3° and 60°, respectively), but after 12:30 on July 22, 2019) the maximum range $R_{K-1}$ was increased to 3000 m ( $h_{K-1}$ of 1734 m and 2600 m at elevation angles of 35.3° and 60°, respectively)." has been added.

16. p. 6, l. 5ff: How is the threshold of -15 dB for SNR determined? How is the relative error of 30 % for turbulence parameters determined? How do clouds and fog affect the measurements in clear parts of the atmosphere? Please explain. Given that the whole time period encompasses 10 days only, it could be interesting to show 10-day time height sections of SNR and wind for the whole period to get an overview and information on the variety of atmospheric conditions.
From measurements on July 28 and 29 (the last two full days of the experiment), we found that above 500 m, the SNR did not exceed -15 dB (this is not a threshold, this is a fact).
Page 6, line 14: "~ 3 km" has been replaced by "2.6 km".
Page 6, lines 14-15: The sentence " Unfortunately, during the lidar measurements on July 26 and 27, there was a series of technical failures (rather lengthy), which made the obtained data unusable." has been added.
Page 6, lines 17-18: The text "(the method for calculating the error is described in papers by Banakh et al.(2017 and 2020))" has been added.
Page 6, lines 20-21: The sentence " Thus, we have data measured by the lidar for 6 days (from 20 to 25 July 2019) and which can be used to determine the heights of the mixing layer." has been added.
Figure 5 has been added.

17. p. 6, l. 12: What is meant by 'the elevation angle was alternated for $\Delta\tau \approx 1.5$s? Until here, I was assuming that consecutive full azimuth scans were done one after the other at each elevation angle.
$\Delta\tau$ is the period of time during which the elevation angle changes from 60° to 35.3° or vice versa.

18. p. 6, l. 23: An objective comparison e.g. by calculating RMSE and correlation coefficient, would be more meaningful than stating that there are practically no differences.
If the difference is negligible, then what's the point of calculating the RMSE and the correlation coefficient. At least for this paper, this is not so important.

19. p. 7, l. 2-3: 'sufficiently high signal-to-noise ratio': how is that determined? What is the criterion to distinguish between sufficiently high and low SNR?

Page 7, Line 16: The reference "(Banakh et al., 2017)" has been added. A description of the calculation of the error in estimating the dissipation rate depending on the SNR is given in this paper. We assume that the SNR is sufficiently high if the relative error in estimating the dissipation rate does not exceed 30%.

In Figures 2 and 3, the maximum heights have been changed from 1400 m to 1800 m (for the elevation angle of 60°) and from 1000 m to 1200 m (for the elevation angle of 35.3°). Page 7, line 12: " We replaced "1000 m" by "1200 m" and "1500 m" by "1800 m".

20. p. 7, l. 4ff: Like above an objective comparison by calculating RMSE and correlation coefficient would be more meaningful. Also, the comparison should be done for the whole period to make the result more meaningful.

The difference in the estimates of the dissipation rate obtained from measurements at different elevation angles is within the error calculated by the algorithm given in the paper of Banakh et al. (2017).

21. p. 7, l. 14-15: This is interesting. Do the authors have any hypothesis why TKE was similar while dissipation rate decreased with height?

This means that the integral scale of turbulence increases monotonically with height. Apparently, the kinetic energy of turbulence remains almost unchanged in the height interval from 60 m to 900 m due to strong convection at 13:00.

22. p. 7, l. 17-18, Fig. 5: The profiles plot do not really show any new information compared to the time height sections. What is the purpose of including them?

In the revised manuscript, we removed Figures 4 and 5.

23. p. 7, l. 22: Where does the threshold used for the radial velocity variance profiles come from? Is that based on other literature? Is it empirical?

For estimation of the MLH from the dissipation rate profiles we the threshold equal to $10^{-4}$ m$^2$/s$^3$. In the same time for estimation of the MLH from the radial velocity variance profiles we the threshold equal to 0.1 m$^2$/s$^2$. According to the calculation using Eq.(1) in the paper by Banakh and Smalikho (2019), at such threshold values ( $\varepsilon = 10^{-4}$ m$^2$/s$^3$ and $\bar{\sigma}_r^2 = 0.1$ m$^2$/s$^2$ ), the integral scale of turbulence $L_v$ is approximately 200 m in the case of lidar measurement at elevation angle of 60°. Such $L_v$ is quite consistent with the results of our measurements in the daytime at heights of 200 - 600 m. Therefore, we used this threshold (0.1 m$^2$/s$^2$ ) for the radial velocity variance.

24. p. 7, l. 25: In case the values of $\sigma_r$ or $\varepsilon$ are smaller than their corresponding threshold, the mixing layer height is set to 60 m. Why not set it too missing? It is well possible that no mixing layer exists at all.

In Figures 6 and 10, we removed the results obtained when the specified threshold exceeds the dissipation rate at an height of 60 m.

25. p. 7, l. 28: similar like to the comment about the sufficiently high SNR above. Which objective criteria is used to determine low quality data?
The data should not contain bad (false) radial velocity estimates. With the accumulation number of 7500 laser shots, this will most likely occur if the signal-to-noise ratio is not lower than -16 dB (provided that the backscatter coefficient is statistically uniform at fixed height). We have determined this SNR threshold from numerical and atmospheric experiments. If the signal-to-noise ratio changes dramatically when the azimuth angle changes, reaching unacceptably low values, then the criterion for determining the suitability of the data becomes significantly more complicated.

26. p. 8, l. 3-4: An objective comparison between mixing layer height estimates for the different elevation angles could easily be done for all available days.
We think that measurements at an elevation angle of 60° are quite optimal and sufficient to obtain information about the height of the mixing layer, especially in the case of large height $h_{mix}$ at which at an elevation angle of 35.3 degrees, the SNR can be unacceptably low.

27. p. 8, l. 8ff: What is the purpose of showing all these examples with plots of dissipation rate and SNR profiles? Like outlined in my major comment, an objective and statistical comparison would make the results more meaningful. The example plots for the individual days could e.g. be put in a supplemental or appendix.
We have removed figures 7-8 (this figures are numbered in the original version of the manuscript).

28. p. 8, l. 17-18: Can this change in SNR be related to a change in wind direction which could explain the enhanced advection of smog?
All day on July 20, 2019, north and northeastern winds were predominantly at the site of the lidar experiment (see figure 5c in the revised manuscript, azimuth angle of 0° corresponds to the direction from north to south). We know for sure that there was a forest fire in the northeast of the experiment site around the same time period. Because of this fire, all of Tomsk was in smog. We do not know when the fire started, but apparently after 18:00 the wind brought smog from the fire to the area of the experiment.

29. p. 8, l. 19-21: How reliable is $\varepsilon$ calculated in the cloud layer? The mixing layer height is determined at the top of the layer with high SNR, i.e. somewhere in the lower part of the cloud. Depending on the cloud characteristics mixing may reach up much further. Thus, the estimated height does not necessarily agree with the true mixing layer height but is simply an affect of how deep the lidar beam penetrates into the cloud.
To estimate the dissipation rate, we used data satisfying the threshold for the signal-to-noise ratio. We did not specifically consider the issue of the accuracy of estimating the dissipation rate in the presence of clouds.

30. p. 9, l. 14: Please explain the difference between probing volume (30 m) and range gate length (18 m).

Our Stream Line lidar emits 170 ns pulses. The range gate length of 18 m corresponds to a 120 ns time window. Then, according to calculations using Eq. (2.34) in the monograph by Banakh and Smalikho (2013), the longitudinal dimension of the probing volume is 30 m.

31. p. 9, l. 17-18: The fact that cloud base and the detected mixing layer coincide does not confirm the correctness of the method. As seen in the examples, mixing layer heights are detected at the top of the layer with maximum SNR, i.e. in the lower part of the cloud. Mixing layer heights may be deeper. The correctness of the method can only be confirmed by comparing it to independent reliable measurements, such as radiosonde profiles.

We fully agree with this statement of the Reviewer. Unfortunately, we have no radiosonde profiles.

32. p. 9, l. 24: It would make it easier to understand how the relative error of TEDR is calculated, if the equation was given.

Page 9, line 11-12: " ( $E_\varepsilon = \left[ < (\hat{\varepsilon}/\varepsilon - 1)^2 > \right]^{1/2} \times 100\%$ , $\hat{\varepsilon}$ is estimate and $\varepsilon$ is true dissipation rate)" has been added.

33. p. 9, l. 26: Why only look at the relative error at mixing layer height? What is the time height section of this error?

Indeed, we obtain the height-time distributions of the error of the radial velocity estimate and the error of the dissipation rate from the data of the atmospheric experiment, but we consider it inappropriate to present such distributions in this paper. Since the accuracy of estimating the MLH is mainly influenced by the error in estimating the dissipation rate at this height, Figure 7 (in the revised manuscript) shows the time series $E_\varepsilon(h_{\text{mix}}(t_{n'}))$ .

34. p. 9, l. 27ff, Fig. 11: In my opinion, Fig. 11 is not ideal and the explanation why the error is high or low could be much easier to follow if a scatter plot between the error and SNR was shown.

According to Figures 8a and 8c (in the revised manuscript), it is possible to obtain the dependence of the relative error in the dissipation rate estimate on the SNR. However, it should be borne in mind that this relative error $E_\varepsilon$ essentially depends on the magnitude of the dissipation rate $\varepsilon$ , that is, $E_\varepsilon$ is a function of SNR and $\varepsilon$ . The paper by Banakh et al. (2017) is devoted to the study of the relative error $E_\varepsilon$ depending on the SNR and $\varepsilon$ .

35. p. 10, l. 21ff: What is the purpose of the series of closed numerical experiments? It is not clear how they are done. What are the preset values of the mixing layer height?

A detailed description of the algorithm for numerical simulation of random realizations for estimating the profiles of the dissipation rate is given in the article by Smalikho and Banakh (2013). It is not the mixing layer height that is important here, but the vertical gradient of the dissipation rate. The main goal of the closed experiments is to estimate the correlation coefficient $C_\xi(l\Delta h)$ , which is then used in the

numerical simulation of random realizations of vertical profiles of the dissipation rate $\hat{\varepsilon}(h_k)$ according to Eq.(6).

36. p. 11, l. 1ff: What is the experimental error? On what assumption is the threshold of 30 % for the relative error based?
Page 10. line 25: "experimental" has been deleted and " using data of atmospheric experiment " has been added. The calculation of the relative error $E_\varepsilon(h_k)$ is carried out with the use Eq.(6) in paper by Banakh et al. (2017). This equation is valid only under the condition $E_\varepsilon(h_k) < 30\%$.

37. p. 11, l. 14ff: While reading this paragraph I was wondering how the calculations are done. This information is given in the following paragraph and I suggest changing the order.
This paragraph has been removed.

38. p. 11, l. 25: What do the random realizations of TEDR look like? How much do they differ from the original profiles? It could be interesting to show some profiles.
Figures 8 and 9 have been added.
Page 11, lines 11-22: The text " Let us consider an example ... = 69 m. " has been added.

39. p. 12, l. 4ff: the error for mixing layer height obtained with the described method is very small. A discussion of other uncertainties related to the mixing layer height, e.g. the sensitivity to the used threshold, should be included. To really assess how correct the determined mixing layer heights are, comparison with independent measurements such as radiosoundings would be necessary. It would be interesting to see if the mixing layer height determined from TEDR would agree with mixing layer heights determined from other instruments (Emeis et al., 2008) within the uncertainty range.
In section 5, the last two paragraphs are replaced by the text " Figure 10 shows ... with SNR of at least -16 dB. " (page 11, lines 23-28; page 12, lines 1-4).
In this experiment, we used only a Stream Line lidar and a temperature profiler (microwave radiometer). We hope that in future experiments we will be able to additionally use other technical means of measurement, including radiosounding.

40. p. 12, l. 22: The Richardson number describes if turbulence can develop in a stably stratified atmosphere (e.g. Stull, 1988). The static stability is described by the temperature gradient.
The Reviewer is right absolutely. This is a slip of a pen. Fixed. Line 7, page 12 of the revised manuscript.

41. p. 13, l. 4ff: It should be mentioned in the beginning that an additional data set from a period in 2020 is used. Also, if data from a microwave radiometer are used much more information on this instrument needs to be given. The temperature profile retrieved from the passive instrument are prone to uncertainties and errors and information on its accuracy and the used retrievals should be given. Microwave radiometers often struggle to resolve elevated inversion at the top of the ABL and thus the gradient Richardson number obtained from these instruments have to be used with care and it should

not be taken as granted that the correctly detect the inversion at the top of the ABL and that they can be used to validate the mixing layer height detected from $\varepsilon$.

We understand that this device measures temperature indirectly and some errors in determining the temperature can arise. We did not study this issue specially. The phrases on page 12, Lines 17-19 of the revised manuscript are added.

42. p. 13, l. 22ff: Like above, a statistical objective analysis of the whole period should be conducted and the conclusion that the mixing layer height derived from $\varepsilon$ agrees well with the gradient Richardson number using a threshold of 0.5 should be based on the whole data set and not just on a single example day.

Text on 13 and 14 pages of the initial version of the manuscript is changed. We extended the experimental data which are considered and analyzed. (Pages 13-14 of the revised manuscript.) 43. p. 14, l. 5: A mixing layer height of several hundred meters during the night must be shear driven. A discussion of the physical processes causing the mixing layer is missing and should be added to the pure description of the profiles.

The phrases on page 14, Lines 11-14 of the revised manuscript are added.

44. p. 14, l. 20: Second period in 2020 should be mentioned.

Fixed, page 15, Lines 16-17 of the revised manuscript.

45. p. 15, l. 13-14: It is not clear to me where the vertical gradient of TEDR is considered in the error estimate.

In calculating the error MLH estimate, we use in equation (6) not the vertical gradient $\gamma$ of the dissipation rate, but the vertical profile of the dissipation rate. We suppressed different values of the vertical gradient $\gamma$ only in numerical experiments. Obviously, the slower the dissipation rate decreases with height, the smaller the vertical gradient $\gamma$ and, as shown by numerical experiments, the larger the error of MLH estimate.

46. p. 15, l. 14: The result 'SNR should be no less than -16 dB' is not clear to me. On what analysis is that based?

The data should not contain bad (false) radial velocity estimates. With the accumulation number of 7500 laser shots, this will most likely occur if the signal-to-noise ratio is not lower than -16 dB (provided that the backscatter coefficient is statistically uniform at fixed height). We have determined this SNR threshold from numerical and atmospheric experiments.

47. p. 15, l. 16ff: A good way of showing this could be by plotting the error of the mixing layer height over $\varepsilon$.

The errors of estimation of the MLH in the experiment in July 2019 are demonstrated in Fig. 10

---

## Referee Report (RR1)

**Round 2: Review of 'Estimation of the height of turbulent mixing layer from data of Doppler lidar measurements using conical scanning by a probe beam' by Banakh et al.**

**December 22, 2020**

The revised version of the manuscript is much improved and the authors considered many of the reviewers' comments. I particularly appreciate that the authors now use more data for their analysis. I still have some comments and suggestions which should be considered, before I can recommend the manuscript to be accepted for publication.

**1 Specific comments**

1. Abstract: I suggest giving the number of days on which the results are based in the abstract. It should be made clear that the comparison of the mixing layer height with the Richardson number estimates are from another measurement campaign.

2. p. 1, l.21-22: Stronger turbulence does not necessarily always lead to a deeper mixing layer height. Other factors such as the inversion strength and vertical motion at the top of the ABL impact the mixing layer height as well (see e.g. Eq. 6.13 in Garratt (1994)). The statement needs to be rephrased.

3. p. 2, l. 5: Specify what $r, u, v, w$ are.

4. p. 2, l. 20: 'MLH $h_{mix}$ is double. One abbreviation is enough. Also see my comment about the usage of abbreviations from round 1.

5. p. 2, l. 30-31: Please specify in the manuscript what is meant by 'sufficiently high SNR'. In the response to my comment from the first review round, the authors explained it nicely: 'We assume that the SNR is sufficiently high if the relative error in estimating the dissipation rate does not exceed 30%.' This statement needs to be included in the manuscript.

6. p. 3, l. 10-12: Like in the abstract, it needs to be made clear that the comparison is from another experiment.

7. p. 4, l. 9: Wind by itself is not a process. The wind field may be stationary.

8. p. 4, l. 20: Add that $D_l$ is a function of $R_k$.

9. p. 5, l. 12-18: The threshold for radial velocity should be given here as well and the reasoning why this thresholds are chosen should be included in the manuscript, like stated in the authors' response to the reviewers' comments in round 1: 'For estimation of the MLH from the dissipation rate profiles we the

threshold equal to 10-4 m2/s3. In the same time for estimation of the MLH from the radial velocity variance profiles we the threshold equal to 0.1 m2/s2. According to the calculation using Eq.(1) in the paper by Banakh and Smalikho (2019), at such threshold values ( $\epsilon = 10^{-4}$ m$^2$/s$^3$ and $\overline{\sigma_r}^2 = 0.1$ m$^2$/s$^2$ ), the integral scale of turbulence $L_V$ is approximately 200 m in the case of lidar measurement at elevation angle of 60°. Such $L_V$ is quite consistent with the results of our measurements in the daytime at heights of 200 - 600 m. Therefore, we used this threshold (0.1 $^2$/s$^2$) for the radial velocity variance.'

10. p. 6, l. 9ff: Refer to Fig. 5 here, as it gives an overview of the atmospheric conditions.

11. p. 6, l. 16-17: 'Estimates of wind turbulence parameters from the data obtained at this SNR [-15 dB] have a relative error exceeding 30%'. Here the authors state that -15 dB are enough. In the abstract and other places in the manuscript they state -16 dB. This needs to be consistent.

12. p. 6, l. 18-20: Give some examples when cloud or fog were present, e.g. morning of July 21, July 22 and July 25.

13. p. 6, l. 25: The explanation what $\Delta\tau$ means is given in the authors' response in round 1. For clarification, this information needs to be included in the manuscript as well.

14. p. 7, l. 14-17: It is confusing to have the same color for missing data and for data outside the colorbar range. Please change.

15. p. 7, l. 27-28: 'The minimum height ...' contradicts with the statement on p. 8, l. 1-2. Also, if MLH at the minimum is not longer given in the plots, this sentence needs to be removed here.

16. p. 8, l. 13ff: Add MLH to time-height sections of Fig. 5, to allow for an easier comparison of conditions and detected MLH. E.g. when describing the MLH minimum at 15 h on July 21.

17. p. 8, l. 16: Although the issue of the accuracy of estimating the dissipation rate in the presence of clouds is not specifically considered in the manuscript, it needs at least to be mentioned that clouds may impact the mixing layer height estimates and the heights strongly depend on how far the lidar beam penetrates into the cloud.

18. p. 8, l. 18: 'changes in the wind with height': What changes? Wind direction, wind speed or both? 'which is apparently the reason': What do the authors mean by that? Is a different air mass advected? Do these changes lead to a decay of turbulence? Maybe rephrase to 'changes in wind direction and speed coincide with a minimum in MLH' or similar.

19. p. 8. l, 19: What is the reason for the low SNR in all layers between noon on July 22 and noon July 24? Is this caused by lidar settings (jump in SNR below 500 m on July 22) or is it physical? In either case, please explain this very prominent feature.

20. p. 8, l. 22, Fig. 6: I recommend to put all six days in one plot? This would make it much easier to compare (like e.g. p. 8, l. 30).

21. p. 8, l. 23: It looks like the highest values on July 20 and 23 occur when the upper most measurement level is taken as mixing layer height. This should be mentioned.

22. p. 8, l. 28-29: The explanation about the difference between probing volume and range gate length given in the authors' response in round 1 should be included in the manuscript.

23. p. 9, l. 1-2: I understand that there are no radiosoundings for verification of the method. However, the agreement between cloud base and mixing layer height is not enough of a justification that the method is correct. The agreement strongly suggests that the method works (in the presented cases),

but it is no proof. The ABL may be deeper. Please rephrase the statement 'The agreement between cloud base and mixing layer height confirms the correctness of the MLH time series assessment.'

24. p. 9, l. 3: It is not clear what is meant by that? Not contradict in what way? Please clarify. How do they compare?

25. p. 10, l. 2: Here and at other places in the manuscript, mix between present and past tense.

26. p. 10, l. 16-17: The purpose of this closed numerical experiments as given in the authors' response in round 1 should be included here.

27. p. 10, l. 25: What is meant by 'atmospheric experiment'? Data from the measurements during the forest fires? Please clarify.

28. p. 13, l. 15, Fig. 11: White curves in Fig. 11 are hard to see. Please change color. I suggest extending the color scale for the gradient Richardson number to negative values, to allow to distinguish between dynamically and statically unstable conditions.

29. p. 13, l. 23-24: The MLH obtained from the gradient Richardson number using the objective threshold method needs to be included in Fig. 11. At the moment it is not clear where this MLH is located and how it relates to MLH from the dissipation rate.

30. p. 13, l. 25ff: The description of MLH April on 10 is unnecessarily detailed. The authors describe the relatives deviations of MLH for different days. This is the first time the talk about relative deviations. I assume the mean the error in the mixing layer estimates $\sigma_h$? The terminology needs to be consistent. The information on MLH uncertainty described in the text is not at all visible in Fig. 11. I highly recommend including the uncertainty in the plots. At the moment it is not clear which periods suffer from a higher uncertainty and which not and it would help to interpret the comparison between MLH from dissipation rate and gradient Richardson number.

31. p. 14 l. 6-7: As MLH from gradient Richardson number is not indicated in Fig. 11, it is not clear how the 22% differences are calculated. In line 25, the authors state relative deviations not exceeding 25%. Which value is correct? Please clarify.

32. p. 14, l. 11ff: Strong wind alone does not lead to low gradient Richardson numbers. It is necessary to have strong wind shear. I cannot follow the examples given by the authors. Between 0 and 6 h on April 10, I see l low gradient Richardson number in the layer between around 250-700 m. Are these the layers with significant shear? Was there a low-level jet? How does that relate to the wind profiles? Same on May 1, I don't see how high wind speed between 150 and 650 m and strong shear between 75 and 200 m links to the observed gradient Richardson number distributions with high values between 200 and 400 m.

33. p. 14, l. 23-24: Add information that data from a second experiment are also used.

34. p. p.14, l. 32: Why does MLH depend on wind? Please clarify.

35. Summary: In their response in round 1, the authors state 'The method described here can be applied to data measured by a high-power pulsed coherent Doppler lidar under normal conditions (with a background aerosol), which will enable a full-fledged statistical analysis.' This is a very valuable information for the reader and I highly recommend including this in the summary or introduction.

36. Fig. 1: Change 'velocity' to 'speed' in label of (f).

37. Fig. 2: I can see differences between both plots, even with this color scale (one color per order of magnitude). Differences might be even be more visible if a finer color scale (like for radial velocity variance in Fig. 3) was used. A difference plot would help to see differences between dissipation rate and radial velocity variance from both elevation angles more clearly and support the statement that the differences for radial velocity variance are larger.

38. Fig. 4: Like in Fig. 6 and 10, the results obtained when the specified threshold exceeds the dissipation rate at an height of 60 m should not be shown.

39. Fig. 5: Change 'wind velocity' to 'wind speed'.

40. Fig. 9: Make clear in the caption that these are 4 examples of random realizations.

**References**

Garratt, J. R.: The atmospheric boundary layer, Cambridge atmospheric and space science series, Cambridge University Press, 316 pp, Cambridge, 1994.

---

## Referee Report (RR2)

The revised manuscript shows that the MLH is successfully retrieved from the TKEDR measured by the CDWL using conical scanning by a probe beam. The relative error of the retrieval results not exceed 20% with the CNR no less than -16 dB. Also, the results agree qualitatively with the MLH derived from Richardson number. Based on my evaluation, I recommend the manuscript for publication on AMT. Some minor corrections in the English are needed.

1. In page 3, line 11, page 14, line 5 and page 14, line 9. "distrbution" should be "distribution".

2. In page 13, line 28. "assesed" should be "assessed".

3. In page 13, line 8. "More over" should be "Moreover".